# ETMR stem-like state and chemo-resistance are supported by perivascular cells at single-cell resolution

Embryonal tumor with multilayered rosettes (ETMR) is a lethal embryonal brain tumor entity. To investigate the intratumoral heterogeneity and cellular communication in the tumor microenvironment (TME), we analyze in this work single-cell RNA sequencing of about 250,000 cells of primary human and murine ETMR, in vitro cultures, and a 3D forebrain organoid model of ETMR, supporting the main findings with immunohistochemistry and spatial transcriptomics of human tumors. We characterize three distinct malignant ETMR subpopulations - RG-like, NProg-like and NB-like - positioned within a putative neurodevelopmental hierarchy. We reveal PDGFRβ⁺ pericytes as key communication partners in the TME, contributing to stem cell signaling through extracellular matrix-mediated interactions with tumor cells. PDGF signaling is upregulated in chemoresistant RG-like cells in vivo and plays a role in recruiting pericytes to ETMR TME by finalizing a signaling cascade which promotes the differentiation of non-malignant radial glia cells, derived from our 3D model, into pericyte-like cells. Selective PDGFR-inhibition blocked the lineage differentiation into pericytes in vitro and reduced the tumor cell population in vivo. Targeting ETMR-pericyte interactions in the TME presents a promising therapeutic approach.

Embryonal tumor with multilayered rosettes (ETMR) is a rare, aggressive brain tumor, with a poor prognosis, mostly found in young children under 3 years[1–3]. Its biology reveals a major role of stem cell signaling dynamics. Histologically, ETMR is represented by multilayered rosettes of neuroepithelial cells and enriched in neuronal stem cell markers, such as WNT5a, CD44, CD133, LIN28A, HES5, and NES[4–7]. In single-cell transcriptomics, distinct ETMR cellular states have been identified, with the presence of neuro-stem cell-like tumor cells[8]. However, the cellular heterogeneity within ETMR, the tumor microenvironment (TME) composition and their implications for tumor progression and resistance to chemotherapy remain unaddressed. Understanding tumor-TME interactions is critical for improving cancer therapy, as cancer progression and resistance are complex processes involving cell-cell and cell-ECM interactions that promote tumor proliferation, immune evasion, angiogenesis, therapy resistance, and metastasis[9–13]. In the TME, brain vascular pericytes (PC) have been

previously reported to exert pro-tumoral functions, as previously reported for glioblastoma[10,14–17]. In health, PC are essential for neurovascular unit integrity and blood-brain barrier formation, maintaining microvascular function from early brain development[15]. Platelet-derived growth factor (PDGF) signaling, crucial for brain vascular formation, is responsible for recruiting PC for angiogenesis in the healthy developing brain and in diseases, along with transforming growth factor-B (TGF-B), vascular endothelial growth factor (VEGF) and angiopoietin[9,15]. In gliomas, pericyte-derived PDGF signaling has been implicated in vascular modulation and paracrine signaling that promotes metastasis[10,14]. Glioblastoma stem cells can be attracted towards endothelial cells and differentiate into PC for vascular formation, indicating cell-cell interactions associated with lineage transitions. In this work, we explore ETMR cell heterogeneity at the single-cell level to understand the mechanisms of chemoresistance. We characterize three distinct malignant subpopulations, named RG-like, NProg-like

✉e-mail: kornelius.kerl@ukmuenster.de

and Nb-like, resembling neurodevelopmental states, each with distinct gene expression profiles and spatial distributions. We show that PDGFRβ⁺ PC from the TME interact with RG-like and NProg-like cells in vascular niches to support self-renewing tumor cells states. We find PDGFR signaling upregulated in chemoresistant ETMR cells in vivo and contributing to the recruitment of pericyte-like cells in ETMR-forebrain organoids (ETMR-FBO) through the lineage conversion of FBO neuronal precursors. Finally, we demonstrate that CP-673154, a PDGFR inhibitor, reduces PC-like populations in ETMR-FBOs and decreases tumor viability in vivo.

## Results

### Single-cell transcriptomics of murine and human ETMRs identified three distinct subpopulations of malignant cells with a predominance of self-renewing states

To understand the cellular diversity of ETMR, we analyzed single-cell and single-nuclei RNA sequencing (scRNA-seq, snRNA-seq, 10x Genomics) of: i.) human ETMR ($n = 11$), ii.) an ETMR genetically engineered mouse model (mETMR-FB, $n = 8$) and control murine brains (mFB, $n = 2$)[18], iii.) ETMR- forebrain organoids ($n = 33$) and iv.) 2D cell cultures ($n = 4$). Findings from these analyses were integrated with immunohistochemistry, multiplex immunofluorescence and spatial transcriptomic (STs) of human samples, in addition to in 3D vitro and in vivo drug assays (Fig. 1A, Supplementary Data 1).

Aiming to understand the cellular diversity of ETMR compared to a normal brain microenvironment at the same developmental stage, we integrated single-cell transcriptomics data from mETMR-FB and mFB at embryonic day 16.5 (E16.5). We applied a t-distributed stochastic neighborhood embedding (t-SNE) for dimensionality reduction and clustering, which effectively separated mETMR-FB and mFB (Fig. 1B). Using cell type-specific marker genes (Supplementary Data 5) and projections onto reference scRNA-seq datasets of murine brains during development[19–21], we grouped the mFB cells into nine major non-malignant cell types (Fig. 1C; Supplementary Fig. S1A).

We identified ETMR cells by a gene signature composed of previously established marker genes of murine ETMR, upregulated downstream of its genetic alterations and discriminating from the background murine forebrain (*Axin2, Smo, Gli1, Gli2*) (Supplementary Fig. S1B)[18]. We further provided a detailed description of the cellular diversity in the murine forebrain (Supplementary Fig. S1C–E), identifying distinct neuronal populations at various stages of differentiation as expected during the course of neurogenesis: i.) radial glia (RG) (*Fabp7, Dbi,* and *Vim*), ii.) neuroprogenitor (NProg) (*Hmgb2, Eomes,* and proliferating markers as *Top2a* and *Ccnd2*) and iii.) early neuroblast (Nb) cells (*Stmn2, Stmn3,* and *Tubb2a*); as well as excitatory neuroblasts (Ex_Nb) (*Neurod2* and *Neurod6*-positive), inhibitory neuroblast (Inh_Nb) (*Gad2, Dlx1, and Dxl2*), and oligodendrocyte progenitor cells (OPCs) (*Olig1* and *Pdgfrα*) (Supplementary Fig. S1E).

In the tumor compartment, murine ETMR was composed of distinct subpopulations of cells resembling transcriptional signatures of RG, NProg, or Nb cells (Fig. 1D), suggesting a putative neurodevelopmental lineage trajectory within the tumoral subpopulations (Supplementary Fig. S2). These trajectories confirmed the expected developmental lineage among the neuronal control cells, with RG cells (cluster 13) transitioning to NProg cells (Clusters 9, 24) and further differentiating to neuroblasts (clusters 3, 6, 12, 15, 17). Strikingly, the ETMR cells followed the same lineage trajectory, with the presence of RG-like (ETMR16), NProg-like (ETMR0, 1, 10, 27) and Nb-like (ETMR2) cell states (Supplementary Fig. S2A). Markers of RG (*Dbi*), NProg (*Hmgn2*), and Nb cells (*Stmn3, Nhlh1*) were differentially expressed in both mFB and mETMR-FB (Supplementary Fig. S2B, C). Notably, the expression of those genes was restricted to the respective cell types along the lineage path of mFB and mETMR-FB cells (Supplementary Fig. S2D–F).

We further uncovered ETMR subpopulations differences in cell cycle phases, metabolism, signaling pathways and transcriptional regulation (Supplementary Figs. S2G, H; S3B–D). RG-like cells in the quiescent state were fueled by fatty acid oxidation (FAO). Distinct metabolic pathways (FAO, glycolysis, and oxidative phosphorylation) were upregulated in the cycling NProg-like cells, while post-mitotic Nb-like cells relied on oxidative phosphorylation for energy intake (Supplementary Fig. S2G, H). This malignant profile recapitulates the metabolic programming of early neuronal states, as previously described[22,23].

Pathways previously described in ETMR[4,7], such as Wnt/β-catenin, SHH, and Hippo transcription factors (Yap, Tead), were enriched in the NProg-like population, while the Nb-like cells exhibited an accumulation of Pi3k, Wnt/Ca+, besides showing a lower expression of the Hippo transcription factors (Supplementary Fig. S3A). We further observed a differential transcriptional regulation among ETMR subpopulations (Supplementary Fig. S3B–D). Interestingly, RG-like and NProg-like cells mainly differed from their healthy murine counterparts by the upregulation of genes associated with cell fate specification and blockage of forebrain neuronal differentiation[24], such as *Ybx1, Zic1, Prox1, Foxo1, Id3* and *Id1*, revealing a pro-self-renewing transcriptional regulation (Supplementary Fig. S3B, C)[25].

To validate some of the main findings, we confirmed the three ETMR subpopulations in a second independent cohort of mETMR-FB (age E18.5, Supplementary Fig. S4) and further analyzed the transcriptional profile of human tissues using scRNA-seq (Supplementary Fig. S5) and snRNA-seq (Fig. 1E, Supplementary Fig. S6) of two independent human tumor cohorts, discriminating malignant from non-malignant cells by a combined approach of copy number variation (CNV) analysis, cluster assignment by ETMR marker genes from bulk RNA-seq[6] and from the mETMR gene signature, thereby defining the hETMR cell populations in every dataset (Supplementary Fig. S5D–F, S6D, E).

In human ETMR, besides the identification of non-malignant cell types in the TME (Fig.1E, Supplementary Fig. S5G, H), we confirmed the presence of the three malignant subpopulations RG-like, NProg-like and Nb-like (Fig. 1F, G, Supplementary Figs. S5H, I; S6F), by comparing the gene expression of human tumor cells with a published scRNA-seq dataset of the human fetal brain at gestational weeks 17, 18[26] (Supplementary Fig. S6F). In addition, we found the same lineage-defining genes of the murine trajectory (*DBI, HMGN2, STMN2*) selectively expressed among the human tumor cells (Supplementary Fig. S5J). In line with the previous findings for the murine tumors, human RG-like, NProg-like and Nb-like were also defined by distinct transcriptional regulators (Supplementary Fig. S5K).

Even though ETMR is composed of cells that resemble the neuronal populations of a developing embryonic brain[27], the relative proportions of the neuronal subpopulations differed from mouse paired healthy tissue (Fig. 1H). While Nb dominated the forebrain of the healthy E16.5 murine embryo, NProg-like cells were the predominant subpopulation in both murine and human ETMR. The prevalence of this tumor subpopulation points to two distinct hypotheses: either a stalled differentiation in ETMR, supported by its transcriptional regulation pattern, or a proliferative/adaptive advantage of the NProg-like cells. For both hypotheses, the maintenance of a stem cell-like state with self-renewing potential appears to be a crucial survival advantage for the tumor.

### Pericytes enrich the extracellular matrix of ETMR with stem cell signaling

As tumor cell differentiation has been directly linked to cellular interaction with the microenvironment and distinct niches in other cancer entities[28–30], we further investigated how ETMR cells interact with cells from their TME to promote pro-tumoral functions. In both murine and human scRNA-seq datasets, the ETMR TME is composed of

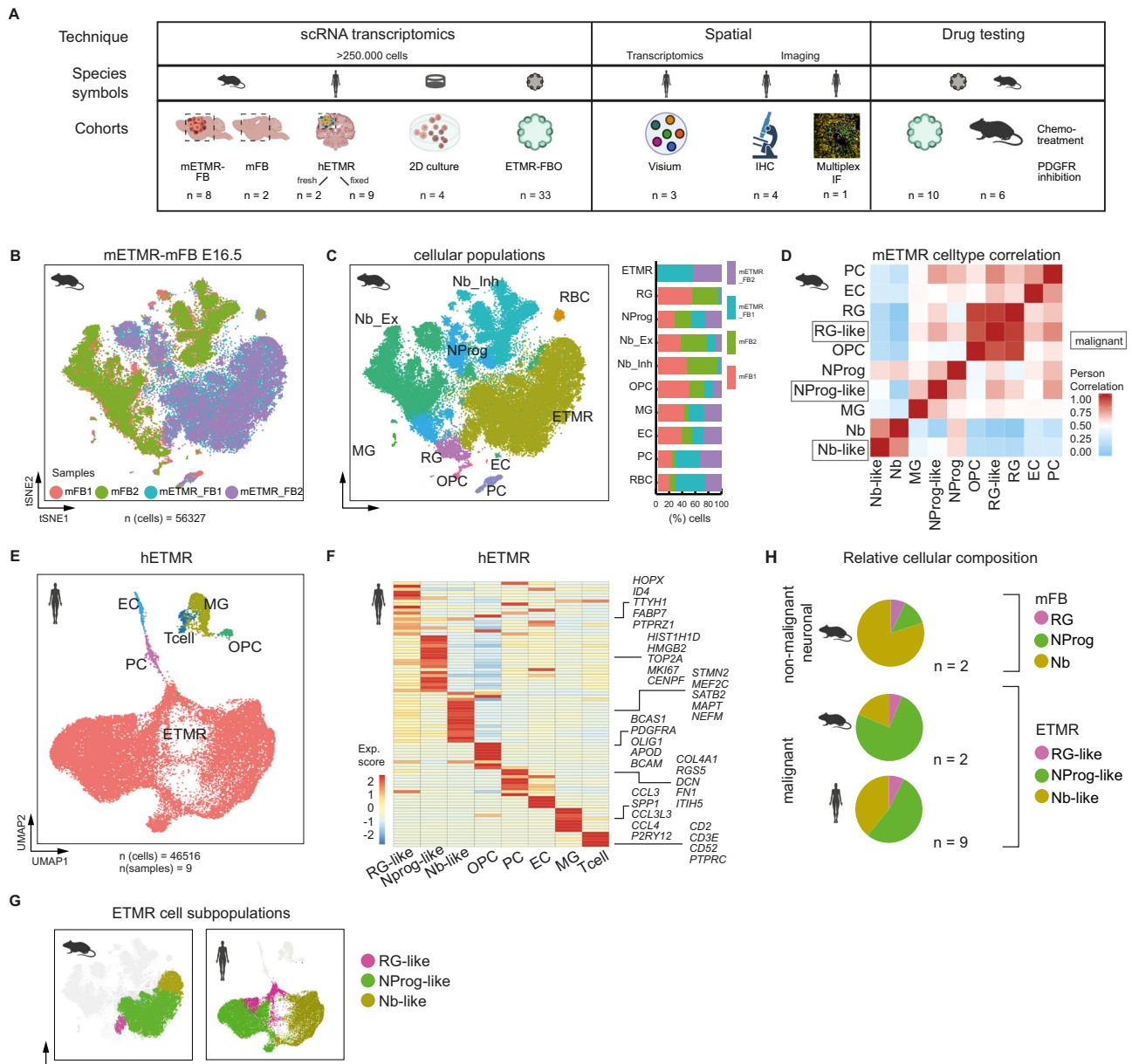

**Fig. 1 | scRNA-seq of murine and human ETMR defined tumor cellular diversity and revealed three distinct ETMR malignant cell states during putative neuronal development. A** Schematic of the experimental design and cohorts. We integrated scRNA-sequencing (scRNA-seq) from over 250,000 cells, including murine forebrains harboring ETMR (mETMR) or not (mFB), human ETMR tumors, 2D cultures and ETMR co-cultured with forebrain organoids (ETMR-FBOs). Quantitative immunohistochemistry (IHC), immunofluorescence (IF), spatial transcriptomics of human ETMR samples and drug testing in ETMR-FBO and tumor-bearing mice were also performed. Species symbols guide related results throughout this study. Partially created with BioRender.com. **B** t-distributed Stochastic Neighbor Embedding (t-SNE) plot shows separate clustering of mETMR-FB (*n* = 2) and wildtype mFB (*n* = 2) samples. Colors indicate sample origin. **C** Cell type composition of integrated murine datasets. Left: t-SNE colored by cell; Right: bar graph by sample. **D** Pearson correlation plots comparing gene expression profiles between malignant and non-malignant cells in the murine dataset. Pearson's

product-moment correlation, *p* value < 0.0001 for RG-like vs RG, NProg-like vs NProg, and Nb-like vs Nb. **E** Uniform Manifold Approximation and Projection (UMAP) of human ETMRs (hETMR; *n* = 9 samples), colored by major cell types. **F** Heatmap showing representative marker gene expression from malignant and non-malignant populations of hETMR tumors. Colored by gene enrichment scores. **G** UMAPs show the three ETMR cellular states – RG-like, NProg-like and Nb-like - in murine (left) and human (right) datasets, colored by cell state. **H** ETMR disrupts early neural cellular fate distribution across species. Pie charts compare fate distributions in healthy mFB controls, mETMR-FB and hETMR conditions, colored by cell type as in (**G**). Non-malignant populations are shaded to match their malignant counterparts for better comparison. Source data of (**C**, **H**) are provided as a Source Data file. RG radial glia, NProg neuroprogenitor, Nb_Ex neuroblast_excitatory, Nb_Inh neuroblast_inhibitory, OPC oligoprogenitor cell, MG microglia, PC pericytes, EC endothelial cells, RBC red blood cells. Sample type symbols in (**B–G**) created with BioRender.com.

OPC (*Olig1* and *Pdgfrα*), MG (by *Fcgr3* and *Trem2*), and PC (by *Pdgfrβ*, *Anxa2*, and *Col1a2*) (Fig. 2A–C). We confirmed the presence of gene signatures of those cell types in bulk-RNA sequencing of 28 patient-derived ETMR samples[31] (Supplementary Fig. S7A). In addition, T cells and endothelial cells (EC) were found in the human snRNA-seq cohort (Fig. 2B).

Next, we characterized ligand-receptor (L_R) interactions among malignant and TME cells using CellPhoneDB[32] and InterCellar[33]. PCs, associated with GO terms such as blood vessel development, extracellular matrix (ECM) organization and PDGF-β receptor signaling, among others[34] (Supplementary Fig. S7B, C), had a high number of interactions in both murine and human TME (Fig. 2D, Supplementary

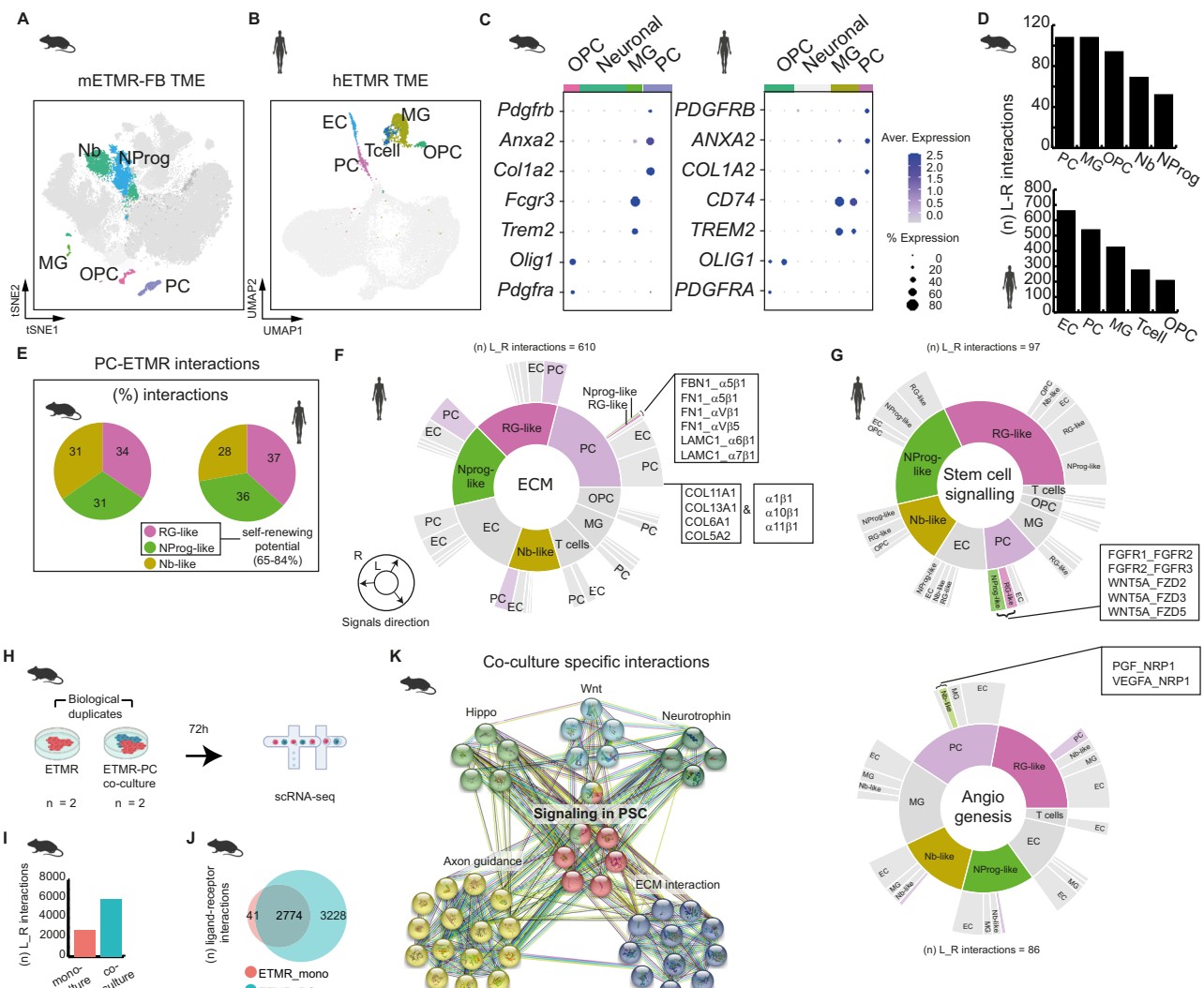

**Fig. 2 | Pericytes are a key player in the TME, influencing tumor ECM and supporting stem-cell signaling. A, B** Representation of the murine (**A**) and human ETMR (**B**) cellular microenvironment in tSNE/UMAP, color-coded by cell type. The remaining cells (malignant and control mFB) are collectively represented in gray. **C** Representative marker gene comparison depicted in dot plots of non-neuronal cell type categories found in both the murine and human datasets. Color-coded by cell category and gene expression level. The dots' size represents the percentage of cells expressing the marker genes. **D** Abundance of cell–cell communications through ligand-receptor (L_R) interactions among TME cells in mETMR-FB (top) and hETMR (bottom). Bar graphs show the total number of interactions (*n*) per cell type. **E** Relative proportion of L_R-interactions from PC to the distinct ETMR subpopulations, depicted in pie charts. **F, G** Sunburst plots representing the interactions in the human snRNA-seq dataset involving ECM (**F**), stem-cell or angiogenesis signaling (**G**). For each plot, the cell types listed in the inner circle represent the signal source, and the cells in the outer circle represent the receiver. The schematic circle with arrows describes the signal direction. Interactions involving ETMR subpopulations and PC are color-coded by cell type; the remaining cell-cell interactions, in gray for clarity. **H** Schematic description of ETMR-PC co-culture experiment. Murine ETMR neurospheres were used in mono- and co-cultures with PCs, with two biological replicates each. Color-coded by cell type: ETMR (red) and PCs (blue). **I** CellPhoneDB analysis of L_R interactions. Bar graph representing the total number of interactions for the two conditions. **J** Venn diagram depicting the number of common and unique (culture condition–specific) interactions in the mono- and co-culture systems. Circles colored by culture condition: mono-culture (red) and co-culture (blue). **K** Co-culture-specific interactions are enriched in pluripotent stem cell (PSC) and neurodevelopmental pathway signaling. STRING network depicting the co-culture unique signaling pathways. L_R molecules are represented by colored spheres grouped by associated signaling pathways. Molecule names omitted for clarity. Souce data of (**D–G**, **I**, and **K**) are provided as a Source Data file. Sample type symbols in (**A–K**) created with BioRender.com.

Fig. S7D). Two-thirds of the outgoing signals (ligands) from the PC cells were mainly directed to tumor cells with self-renewing potential (RG-like and NProg-like) in both murine and human microenvironments (Fig. 2E), raising the hypothesis that PCs could be related to stem cell signaling in ETMR.

As stem cell signaling is a hallmark of ETMR molecular biology, we followed the hypothesis that PC might contribute to this fundamental characteristic of the tumor. Aiming to understand the underlying mechanisms, we investigated the biological role of interactions between PCs and the tumor cells. Firstly, we selected the tumor-specific interactions (interactions present in the mETMR-FB but absent in the mFB), and classified them into three biological categories based on STRING network predictions[35]: i) ECM organization, ii) stem cell signaling, and iii) proliferation/angiogenesis (Supplementary Fig. S8). We validated the findings in a second independent cohort of three mETMR-FB (Supplementary Fig. S4C–E).

We observed that PC participated in interactions of the three main categories, involving *Fn1* and *Tnc_α8β1* integrin, *Cxcl12_Cxcr4*, *Spp1_Cd44*, *Bmp7_Bmpr*, and *Ntf3_Ntrk2*, respectively, pointing to PC multiple roles in the ETMR TME, frequently involving components of the ECM. Notably, *Fn1*, β−integrin (*Itgb1*), *Cxcr4*, and *Tnc* have been previously associated with the ECM of cortical germinal zones in

human and murine, forming a niche that favors the self-renewal of progenitor cells[35,36]. PC were also directly associated with tumor-specific interactions related to stem cell signaling through *Bmp7_Bmpr*, and proliferation and angiogenesis through *Ntf3_Ntrk2* (Supplementary Fig. S8B).

Secondly, we confirmed the contribution of PC for significant interactions within the tumor-specific categories "ECM", "Stem cell signaling" and "Angiogenesis" in the human dataset (n = 9 samples) (Fig. 2F, G). Although all the tumor subpopulations participate in the three categories of tumor-specific signaling, PC engages with RG-like and NProg-like tumor cells mostly through ECM components and participates with them, but not with NB-like cells, in "stem cell signaling".

Thirdly, to verify the influence of PC on ETMR stem cell state, we performed scRNA-seq of murine ETMR neurospheres after 72 hours in mono-culture, and co-culture with PC (Fig. 2H). Tumor cells from the co-culture condition clustered separately from those in mono-culture, indicating the influence of PC on gene expression of specific ETMR subpopulations (Supplementary Fig. S9). In addition, we detected a more than two-fold increase in the number of L_R interaction pairs in the co-culture condition (Fig. 2I), besides a high number of unique interactions (Fig. 2J). Among the co-culture unique interactions, we confirmed the presence of L_R pairs associated with signaling in pluripotent stem cells and ECM network, besides neurotrophic signaling (Wnt, Hippo, neurotrophin) also associated to stem cell niches, as previously described[37], corroborating the findings of the scRNA-seq of primary murine ETMR (Fig. 2K). We did not find a specific angiogenic network, suggesting that additional microenvironment components are necessary for PC to exert angiogenic functions.

## Pericytes co-localize with spatially organized self-renewing RG-like and NProg-like cells

Hypothesizing that the intercellular communication between PC and self-renewing tumor cells requires co-localization within tumor regions, we examined the spatial arrangement of ETMR. Initially, immunohistochemistry (IHC) of human ETMR confirmed previous observations[7] that hypercellular areas were strongly positive for the NSC marker Nestin (NES) and NSC/NProg– marker SOX2, but showed weak synaptophysin expression and lacked NeuN-positive cells (Supplementary Fig. S10A). In contrast, hypocellular areas exhibited sparse NES and SOX2 expression, but were positive for synaptophysin, MAP2c and included some NeuN-positive cells, indicating a higher degree of neuronal maturation.

Based on these IHC profiles, we performed multiplex IF (n sample = 1) and STs (n sample = 3, Visium platform, 10x Genomics) on pathologically assigned primary human ETMR sections, mapping tumor subpopulations (Supplementary Fig. 11A–D). We identified tumor RG-like (NES^{high}; SOX2^+), NProg-like (NES^{low/–}; SOX2^+) and Nb-like (SOX2^–; MAP2^+) cells in protein staining (Supplementary Fig. S11A).

Transcriptionally, the three tumor subpopulations defined three distinct spatial regions based on cell-type enrichment from single-cell integration analysis (Supplementary Fig. S11E). *LIN28A* expression, a histological marker of ETMR, correlated with NProg-like enriched regions, alongside *MKi67* expression. (Supplementary Fig. S11A, F)[6]. PC and EC mostly correlated with spatial regions enriched in RG-like cells (Supplementary Fig. S11G). In addition, we observed that ECM ligands and receptors predicted in the human snRNA-seq cohort (see Fig. 2F, 2G) spatially overlap with PC- and/or RG-like-enriched regions (Supplementary Fig. S11H), supporting ECM-mediated communication between these cell types.

Next, we mapped the spatial localization of pericytes (PDGFRβ^+) in single-cell resolution, confirming their co-localization with vascular areas (Fig. 3A). Notably, we found NES^+ and SOX2^+ tumor cells in closer proximity to PDGFRβ^+ regions, in comparison to MAP2^+ cells (Fig. 3B, C). Quantitative IHC (n samples = 4; fields of view = 325)

revealed significant enrichment of CD13^+ brain PC[38] in hypercellular regions compared to hypocellular regions, confirming the association of PC with tumor areas enriched with RG-like and NProg-like tumor cells (Fig. 3D, Supplementary Fig. S12A). We also observed the differential expression of key PC markers, including *RGS5*, *PDGFRβ*, *COL1A2*, *ACTA2*, *TACLN* and *COL3A1*, in rosette areas (hypercellular) versus neuropil (hypocellular) tumor areas of a published RNA-seq dataset of microdissected ETMR (n sample = 1) (Supplementary Fig. S12B)[7].

## ETMR-forebrain organoids recapitulate ETMR intratumoral heterogeneity defined by scRNA-seq

To better understand the mechanisms of the cellular interactions of ETMR tumor cells with their environmental niche, we developed a 3D culture system by integrating tumor cells into forebrain organoids (FBO) derived from human induced pluripotent stem cells (hiPSCs). We chose FBO, as this reflects the frequent origin of ETMR in the human cortical region[6,18,39]. Thirty days after co-aggregating hiPSCs with either human (BT-183) or murine (GEMM-derived) ETMR cell lines, we analyzed the resulting organoids (hETMR-FBO and mETMR-FBO, respectively) using whole mount immunostaining, histology, and scRNA-seq (Fig. 4A; see Methods). The FBO were uniformly round, similar in size, and contained MAP2^+ neurons (Fig. 4B). Co-aggregation with YFP-labeled murine ETMR cells resulted in mETMR-FBO forming exophytic tumor conglomerates, distinguishing them from the tumor-free FBO (Fig. 4C). Tumor cells in mETMR-FBO (NES^+, some expressing nuclear β-Catenin) and hETMR-FBO (NES^+; LIN28a^+) recapitulated the major histopathological features of primary ETMR, including a similar neuronal differentiation profile (Fig. 4D; Supplementary Fig. S13A, B).

Further, we characterized the cellular composition of ETMR-FBO by integrating scRNA-seq datasets of one FBO sample (8-pooled organoids, tumor-free) and two mETMR-FBO samples (8-pooled organoids each) at 30 days after co-aggregation (Fig. 4E). Analysis identified 29 clusters, with malignant ETMR cells forming clusters separate from the FBO cells (Fig. 4E; Supplementary Fig. S13C). Non-malignant cells were grouped into metaclusters representing RG, NProg, and Nb lineages (Fig. 4F; Supplementary Fig. S13D), with RG and Nb further subdivided into subpopulations (Supplementary Fig. S13E, F).

To assess whether ETMR heterogeneity was maintained in organoids, we annotated mETMR-FBO clusters using gene signatures from primary murine tumors. All three ETMR subpopulations were recapitulated, mirroring their proportional representation in primary tumors, unlike neurosphere cultures (Fig. 4G, H; Supplementary Fig. S13G). Similarly, hETMR-FBO preserved cellular heterogeneity (Fig. 4I, K; Supplementary Fig. S14). The loss of heterogeneity in neurosphere cultures underscores the importance of cellular interactions with neural populations in maintaining ETMR diversity in the 3D model.

## ETMR induces a lineage transition from non-malignant RG-to-PC via a WNT-VEGF-PDGFR pathway

We next explored interactions between tumor and FBO cells, identifying non-malignant RG subpopulations that differed between control FBO and mETMR-FBO. In the mETMR-FBO, two unique RG-FBO clusters, termed RG-PC and RG-angio, emerged based on their association with either ECM organization−resembling PC GO terms (RG-PC), or blood vessel development (RG-angio) (Supplementary Figs. S7B, S14E, S15A, B).

These non-malignant RG subpopulations, while not expected in FBOs[40], were present in tumor-co-aggregated FBOs (Fig. 5A, B, Supplementary Fig. S15C). Notably, the emergence of PC-like gene expression in RG-PC cells suggests a tumor-induced lineage shift toward a pericyte-related state. Applying the PC gene signature from primary murine and human scRNA-seq revealed that RG-PC cells shared overlapping transcriptional profiles with PC primary cells

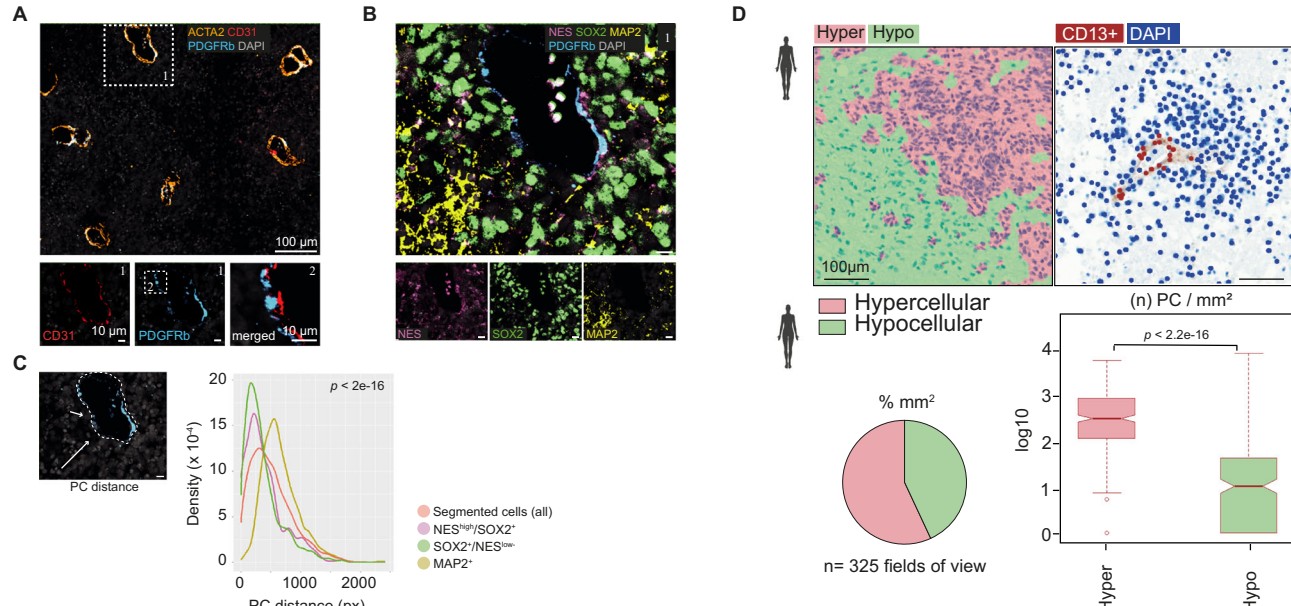

**Fig. 3 | PCs are enriched in ETMR hypercellular regions, which correlate with stem-like cell fates. A, B** Muliplex IF analysis of a human ETMR FFPE section (*n* = 1). **A** Representative multiplex IF images showing merged channels for smooth-muscle cells (ACTA2+), PC (PDGFRb+), EC (CD31+) and DAPI nulear staining. Insets present individual channels of selected areas (dashed boxes 1 and 2), revealing the close juxtaposition of PC to EC in vessel walls (box 2). **B** Amplified image of the region within box 1 (panel A), depicting RG-like cells (NEShigh/SOX2+), NProg-like cells (SOX2+/NESlow−), Nb-like cells (MAP2+) and PC (PDGFRb+). The main panel shows merged channels, while insets display selected single channels (scale bar, 10 µm). **C** Quantitative analysis of tumor subpopulation proximities to PC regions described in (**A**). PC regions were defined by the areas enclosed by PDGFRb+ signal, as illustrated by the dashed line. Arrows indicate the shortest distances of individual cells to the nearest PC region. Distances were measured in pixels (px), and the distributions of each cell type and all segmented cells are shown in a histogram.

NEShigh and SOX2 + /NESlow cells were significantly closer to PC regions than MAP2+ cells (*n* cells = 19,477; One-way Anova, F(3, 43917) = 1256, *p* value < 2e-16; post-hoc test TukeyHSD, *p* value < 0.00000001). **D** Quantitative IHC of human ETMR tumors. The left panel shows representative regions of interest (ROIs) of ETMR IHC sections as hypercellular (Hyper, pink) and hypocellular (Hypo, green) regions (also Suppl. Fig. S12). The right panel depicts DAPI+ cells (blue) and CD13 + PC (red) for further quantification (*n* samples = 4; field of view = 669 µm × 500 µm; *n* fields of view = 325). Scale bar = 100 µm. Boxplot shows median (center line), first and third quartile (bounds) and minima/maxima (whiskers) of all measurements of PC density in the Hyper vs Hypo regions. Data points 1.5× more than the interquartile range away from the box were considered outliers. One-sided Wilcoxon rank sum test with continuity correction. Sample type symbol created with BioRender.com. Data source of (**D**) is provided as a Source Data file.

(Supplementary Fig. S15D, E). Consistent with this phenotype, RG-PC cells showed enrichment in pathways critical for PC differentiation and expansion, including VEGF, TGFβ, and PDGFR, along with ECM pathways such as β-integrin signaling, smooth muscle activity and IGF signaling (Supplementary Fig. S15F)[41,42].

To trace the differentiation path leading to the development of RG-PC cells in the mETMR-FBO, we performed pseudotime trajectory analysis of the non-malignant cells using Monocle 3.0 (Fig. 5Ci, Supplementary Fig. S16Ai). This revealed distinct RG programs driving astroglial, neuronal, or mesenchymal fates (Supplementary Fig. S16B–D). In the RG-to-PC transition, cycling neuroepithelial RG-FBO clusters (*cyNESC*) first adopt a mesenchymal phenotype (*MES*) before specifying into an *RG-PC* program (Fig. 5D). The *cyNESC* program was enriched with mitosis-associated (*CENPF, NUSAP1*) and cortical progenitor (*HMGB2*) genes[19,43]; the *MES* program featured genes associated with epithelial-to-mesenchymal transition (*GJA1, TPBG* and *MDK*)[44–46]; and the *RG-PC* program expressed genes involved in PC and smooth muscle cell development (*PDGFRβ, LGALS1, MGP*) (Fig. 5E)[47,48].

A signaling gradient guided the RG-to-PC transition. Early-stage RG cells from the FBO in the *cyNESC* state downregulated WNT and upregulated Notch signaling, progressing to an intermediate *MES* state with Hippo pathway activity (Fig. 5F)[49]. As the transition progressed, SHH signaling increased, followed by activation of VEGF, TGFβ, and PDGFR pathways, culminating in PC fate determination (Fig. 5F, G).

To elucidate the cell-cell interactions in RG-to-PC differentiation, we examined the L_R pairs between tumor and RG-FBO along the PC lineage. ETMR-specific *WNT5A* ligands interacted with *FZD5* receptors

in *MES*-state RG-FBO cells, potentially guiding them towards a mesenchymal stem cell state (Fig. 5H, Supplementary Fig. S17)[50]. *MES* cells expressed *DLK1*, a putative mesenchymal marker (Kumar, 2019), which activated *NOTCH2* and *NOTCH3* receptors in *MES* and RG-PC cells. Tumor-derived *VEGFA* and *TGFβ2* signals were received by *NRP2* and *TGFβR2/3* receptors present on *MES* and RG-PC cells, respectively, with PDGFA_PDGFRA signaling from tumor cells directing final PC specification.

## PDGFR signaling is harnessed by RG-like tumor cells to evade chemotherapy-induced cell death in vitro and in vivo

Next, investigated the potential implications of PC and PDGFR signaling for the response of ETMR subpopulations to chemotherapy. Firstly, we treated mETMR-FBO and hETMR-FBO models with etoposide (1 µM/day) over a 4-day period. Organoids were harvested at distinct time points representing short (1 day), intermediate (intermed, 6 days) and late (18 days) intervals following the end of the treatment for single-cell transcriptomics. The resulting scRNA-seq datasets from mETMR-FBO and hETMR-FBO were analyzed (Supplementary Fig. S18A–H) and integrated to perform cell-type enrichment analysis across the distinct treatment intervals (Fig. 6A, Supplementary Fig. S18I, J).

In the short interval, we observed a proportional increase in Nb-like gene expression, likely reflecting the expected lower sensitivity of non-cycling cells to chemotherapy. At the intermediate interval, during which surviving cells had the opportunity to emerge post-chemotoxicity, there was a notable enrichment in the expression of RG-like cells, alongside a modest increase in NProg-like cells and a decrease of

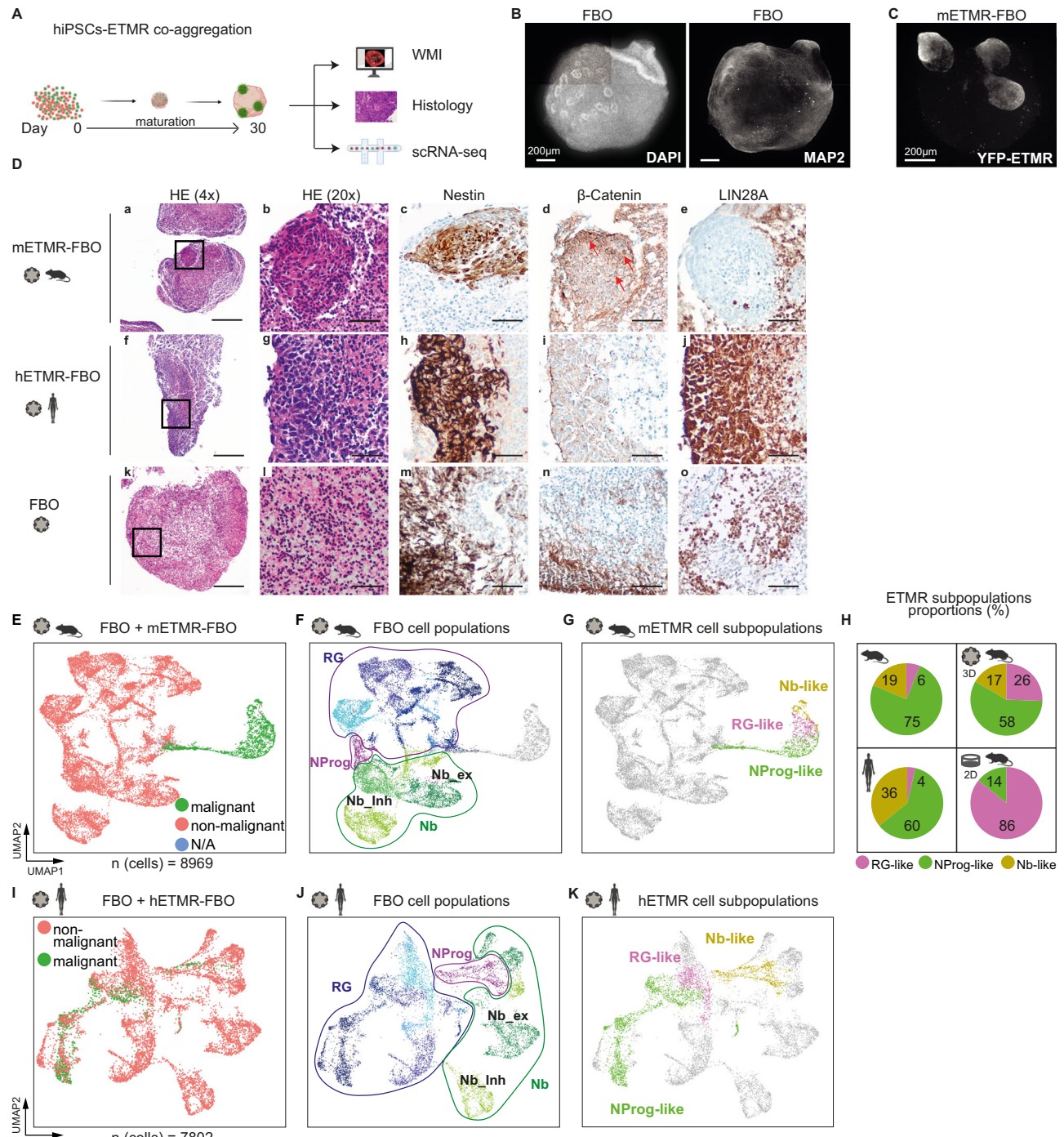

**Fig. 4 | ETMR cellular heterogeneity was recapitulated in ETMR-forebrain organoids. A** Schematic of ETMR-FBO generation via co-aggregation of hiPSCs with murine or human ETMR cells: i) on day 0, hiPSCs were co-aggregated with ETMR cells; ii) organoids were matured under standard conditions; iii) on day 30, ETMR-harboring (ETMR-FBO) and non-harboring (FBO) organoids were used for histology, immunostaining, and scRNA-seq. Created with BioRender.com. **B** Representative maximum projection image of all confocal planes from whole-mount immunos-tained (WMI) FBOs for DAPI and MAP2 (neuronal maker) (*n* = 6 organoids; 1 experiment). Scale bar = 200 μm. **C** Representative WMI image of YFP+ murine ETMR cells integrated in FBOs (mETMR-FBO) (*n* = 36 organoids; 2 experiments). Scale bar = 200 μm. **D** (a–o) Immunohistochemistry (IHC) of mETMR-FBO, hETMR-FBO, and FBO (1x FFPE block with ~ 20 organoids/condition; 2 experiments); **a, f, k**) H&E staining of each condition (×4 magnification, scale bar = 250 μm). Squares indicate areas further characterized by IHC; (**a–e**). Tumoral regions in mETMR-FBO showed

high cell density, Nestin positivity, β-Catenin-accumulation (red arrows) and LIN28A negativity[18]; (f-j), tumor areas in hETMR-FBO showed Nestin and LIN28A positivity (×20 magnification, scale bar = 50 μm). **E–K** scRNA-seq analysis of murine and human ETMR-FBO. **E** Integrated UMAP clustering from one FBO and two mETMR-FBO, showing malignant (murine), non-malignant (human) and non-assigned (NA) cells. **F** UMAP showing main FBO cell types. Colored by cell types (RG radial glia, NProg neuroprogenitor, Nb neuroblast, Nb_ex Nb_excitatory, Nb_Inh, Nb_inhibitory). **G** UMAP of mETMR-FBO showing ETMR subpopulations (RG-like, NProg-like, and Nb-like), colored by cell type. **H** Pie charts of ETMR subpopulation proportions across datasets (primary murine/human ETMR, mETMR-FBO, and 2D ETMR culture). Same color code as in (**G**). **I–K** UMAPs from integrated clustering of FBO and hETMR-FBO (*n* = 1 sample each), showing hETMR and FBO cells (**I**), FBO cell types (**J**), and hETMR subpopulations (**K**) recapitulated within FBOs. Source data of **H** is provided as a Source Data file. Sample type symbol in **D**–**K** created with BioRender.com.

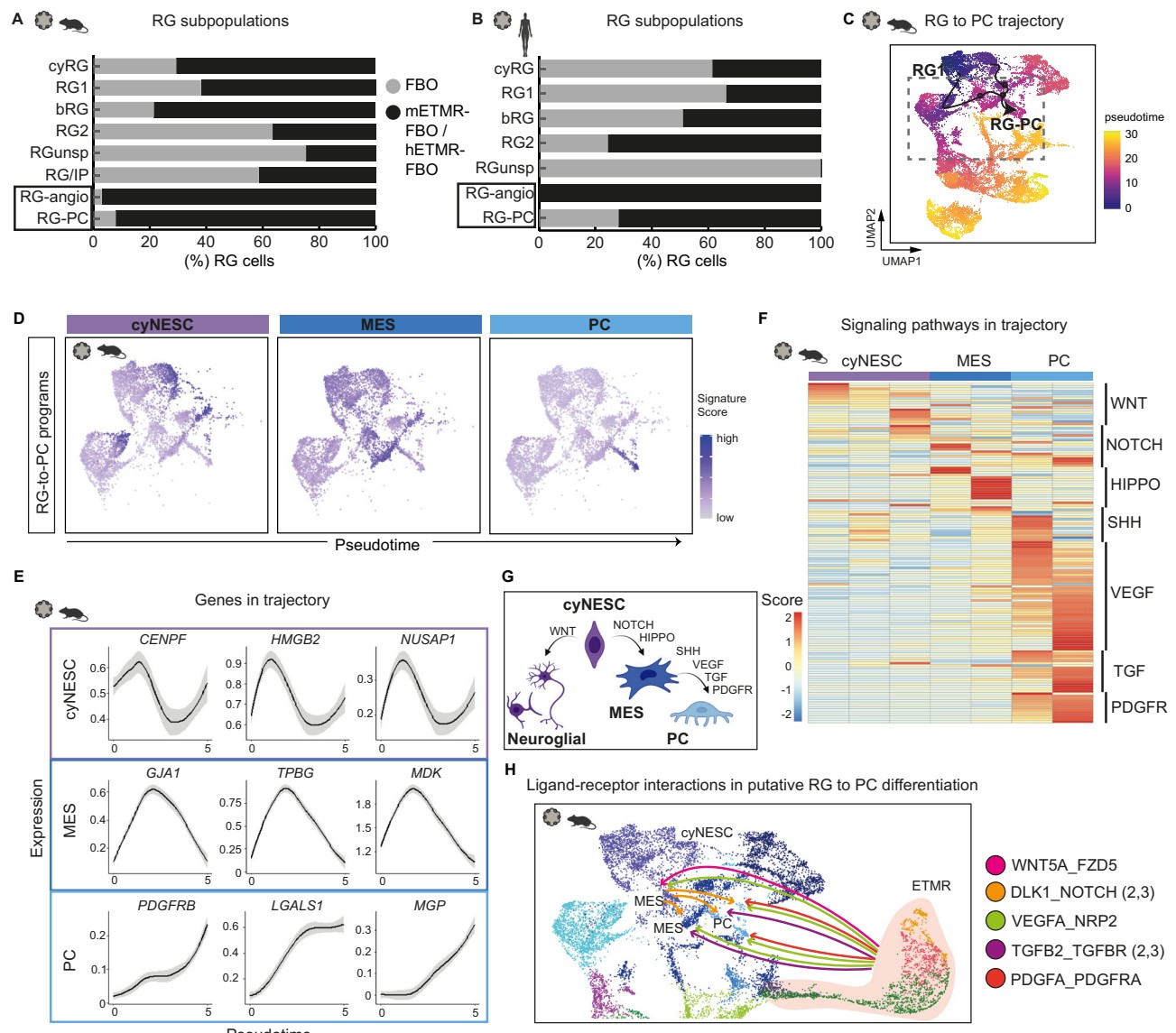

**Fig. 5 | ETMR altered the fate determination of RG subpopulations in the FBO, supporting their differentiation into pericytes through WNT-TGFB-PDGFR signaling. A, B** ETMR altered the fate determination of non-malignant RG subpopulations in either the mETMR-FBO (**A**) or hETMR-FBO (**B**) compared to the FBO controls. RG clusters enriched in vascular development GO terms are referred to as RG-angio, while those enriched in PC marker genes are termed RG-PC. Bar graphs are color-coded by condition. **C** Monocle pseudotime trajectory of RG cells from the mETMR-FBO dataset illustrates differentiation path from RG-to-PC. Cells are color-coded by pseudotime. Arrows highlight the trajectory from early RG subpopulations to RG-PC. Dashed lines mark the clusters included in the trajectory analysis described in (**D**−**G**). **D** UMAP signature plots of transitional programs involved in RG-to-PC trajectory show that RG cells in the cyNESC program first undergo a mesenchymal (MES) transition, which further progress into a PC differentiation program. **E** Line plots illustrate gene expression trends across pseudotime for selected genes differentially expressed along the trajectory. Genes are grouped into functional programs, indicated by boxed sections that mark their association with specific pseudotime stages. Lines represent non-linear smoothed gene expression trajectories (LOESS fit); black lines indicate the estimated central tendency, and gray shaded areas denote the 95% confidence interval based on standard error. **F** Heatmap shows expression scores of marker genes representing signaling pathways that are differentially upregulated in RG clusters involved in the RG-to-PC trajectory. Pathways were identified via ToppGene analysis. **G** Schematic of cell types and signaling pathways involved in RG lineage conversion to PC. Created with BioRender.com. **H** Diagram of predicted L_R interactions (from Cell-PhoneDB/InterCellar) involved in promoting RG-to-PC differentiation (see also Supplementary Fig. S17). Source data of (**A**, **B**) are provided as a Source Data file. Sample type symbols in (**A**−**F**, **H**) created with BioRender.com.

Nb-like cells in comparison to untreated D30 organoid controls (Fig. 6B). This suggests that RG-like cells are the best suited to survive chemotherapy-induced cell injury and, therefore, are associated with chemotherapy resistance. Similarly, treatment with carboplatin (10 μM/day) under identical conditions and time points produced comparable outcomes (Supplementary Fig. S18I). At the late interval, the overall tumor cell gene expression levels remained modestly reduced relative to pre-treated controls at the same time-point (Supplementary Fig. S18J).

Next, to verify the presence of a chemo-resistant subpopulation in ETMR, we treated six ETMR murine embryos at E14.5 either with etoposide at 5 mg/kg/d ($n = 3$) or vehicle ($n = 3$) for three consecutive days and processed the samples for scRNA-seq analysis on E18.5 (Fig. 6C, Supplementary Fig. S19A). After quality control, we selected 12,447 cells for further analysis, revealing balanced distribution among distinct clusters (Supplementary Fig. S19B, C). Cell annotation identified 11 cell types, including ETMR subpopulations (Supplementary Fig. S19D−F), which we focused on for investigating chemo-resistance.

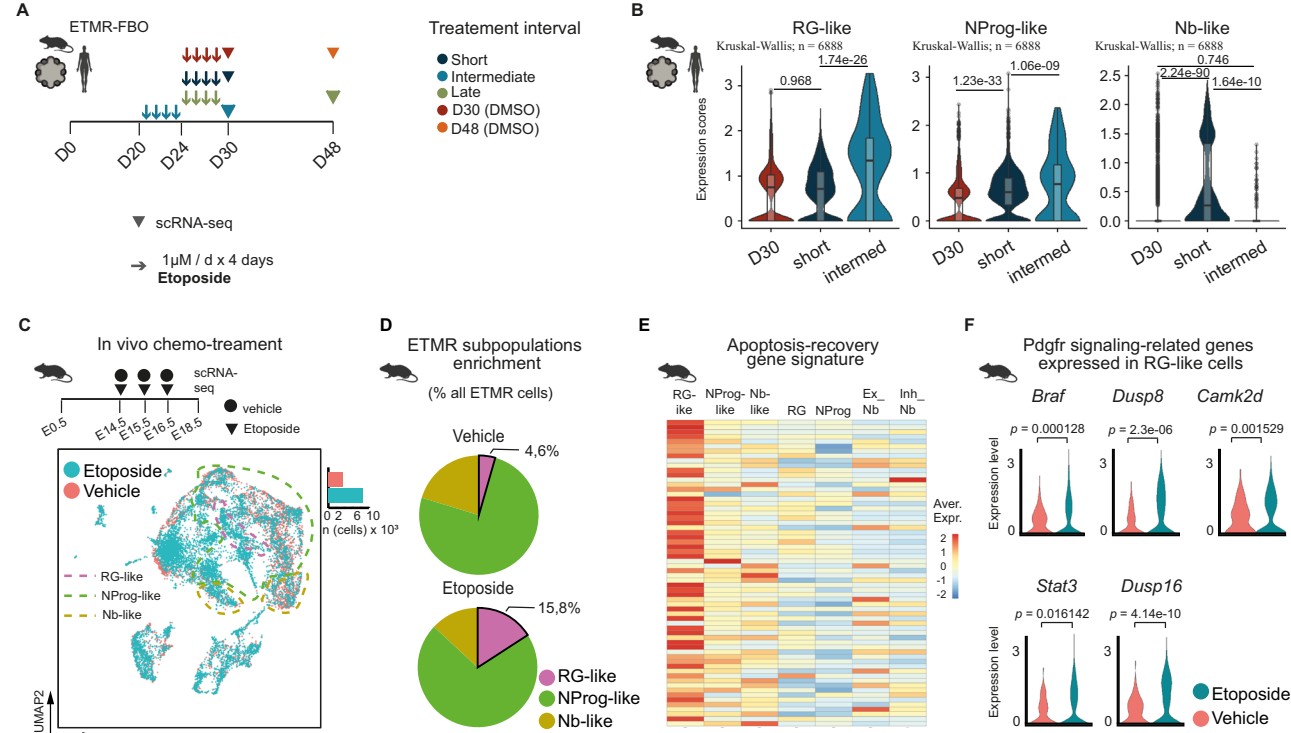

**Fig. 6 | Chemotherapy resistance in ETMR is mediated by RG-like tumor subpopulations upregulating PDGFR signaling. A** Chemotherapy treatment schedule of mETMR-FBO and hETMR-FBOs (2 experiments; ETMR-FBO = 2 biological replicates/treatment-interval; FBO control = 1). Organoids were exposed to etoposide (1 µM) or DMSO as a control. They were treated over a 4-day period and subsequently harvested for scRNA-seq at organoid culture day 30 (D30) or day 48 (D48), after a short (1 day), intermediate (6 days), or late (18 days) interval post-treatment. **B** Expression scores of ETMR cell type-specific gene signatures in integrated scRNA-seq dataset of mETMR- and hETMR-FBO experiments from (**A**), comparing D30 (DMSO control), short and intermediate intervals. We used Kruskal–Wallis statistical test for multi-group comparisons and Wilcoxon rank sum test, two-sided, for inter-group comparisons, with Benjamini–Hochberg false discovery rate (FDR) *p* value adjustment. The number (*n*) of cells for analysis is displayed. **C** scRNA-seq of E18.5 ETMR-harboring murine forebrains (mETMR-FB) treated daily with etoposide (5 mg/kg) or vehicle at days E14.5-E16.5 by i.p. injection in pregnant mothers with *n* (embryos) = 3 per condition (vehicle or etoposide). UMAP describes cellular distribution, color-coded by treatment condition. Dashed lines delineate ETMR subpopulations. **D** Pie chart showing the relative proportion of each ETMR subpopulation in the conditions vehicle versus etoposide-treated samples. Color-coded by ETMR subpopulations. **E** Heatmap depicting the average expression of marker genes representing an "apoptosis-recovery" gene signature[53]. **F** Violin plots depicting Pdgfr-related differentially expressed genes between etoposide versus vehicle-treated RG-like cells, calculated using two-sided MAST statistical test; adj.*p* values are displayed. Source data of **D** are provided as a Source Data file. Sample type symbols in **A**–**F** created with BioRender.com.

We observed an enrichment of RG-like cells in etoposide-treated samples compared to vehicle-treated ones (4.6% versus 15.9%, respectively), supporting the hypothesis that RG-like cells are linked to chemoresistance (Fig. 6D).

To test this hypothesis, we focused on the well-known property of etoposide to induce apoptosis in cancer cells[51]. We analyzed our dataset for a molecular signature associated with genes that enable cells to recover from apoptotic stimuli[52]. Our findings revealed that RG-like cells exhibited the highest enrichment of apoptosis-recovery genes (Fig. 6E), which were significantly more expressed in post-etoposide-treated samples compared to pre-treated ones (Supplementary Fig. S19G). This suggests that RG-like cells have developed adaptive responses to treatment, supporting them as drug-resistant tumor subpopulations.

We further analyzed up-regulated signaling pathways in RG-like cells to understand the molecular mechanisms of chemoresistance. PDGFR signaling emerged as one of the most significantly upregulated pathways in etoposide-treated RG-like cells (Fig. 6F), alongside TGFβ, VEGF and ECM signaling (Supplementary Fig. S19H). Interestingly, PDGF signaling upregulation has been associated with the suppression of miRNA Let-7 family, also implicated in EMTR miRNA dysregulation[53,54]. PDGFR signaling likely upregulates through JAK/STAT3 and CAMK2D pathways activation, as indicated by the gene expression profile (Fig. 6F). Using the KEGG pathways database as a reference [55], we inferred that *JAK/STAT3* might enhance VEGF signaling, while *CAMK2D* may induce transcription of c-Fos and c-Jun targets downstream of BRAF, both potentially contributing to angiogenic effects (Supplementary Fig. S19I).

## PDGFR inhibition blocks PC development in FBO and decreases tumor burden in vivo

Our findings established that PC are central to the ETMR microenvironment. PC are recruited into the TME via PDGFR signaling in ETMR-FBO, engage in stem cell signaling with tumor cells, and exhibit PDGFR signaling patterns associated with chemo-resistant RG-like cells. We therefore evaluated the PDGFR inhibitor CP-673451 for its potential to reduce ETMR cell survival both in vitro and in vivo.

We treated mETMR-FBO with the CP-673451 or vehicle (DMSO) for 15 consecutive days, up to day 30 of organoid culture, followed by scRNA-seq of CP-673451- and DMSO-treated samples. A total of 42,206 quality-filtered cells were annotated according to gene expression profiles (Fig. 7A, B; Supplementary Fig. S20). PDGFR inhibition with CP-673451 resulted in a 10-fold decrease of the PC relative proportions in the TME, although not statistically significant (median: 0.44% in DMSO-treated versus 0.04% in CP-treated samples, *p* value = 0.42) (Fig. 7C). In addition, the treatment reduced ETMR tumor cell numbers (median: 3.87% in DMSO-treated vs. 1.29% in CP-673451-treated samples, *p* value = 0.2)

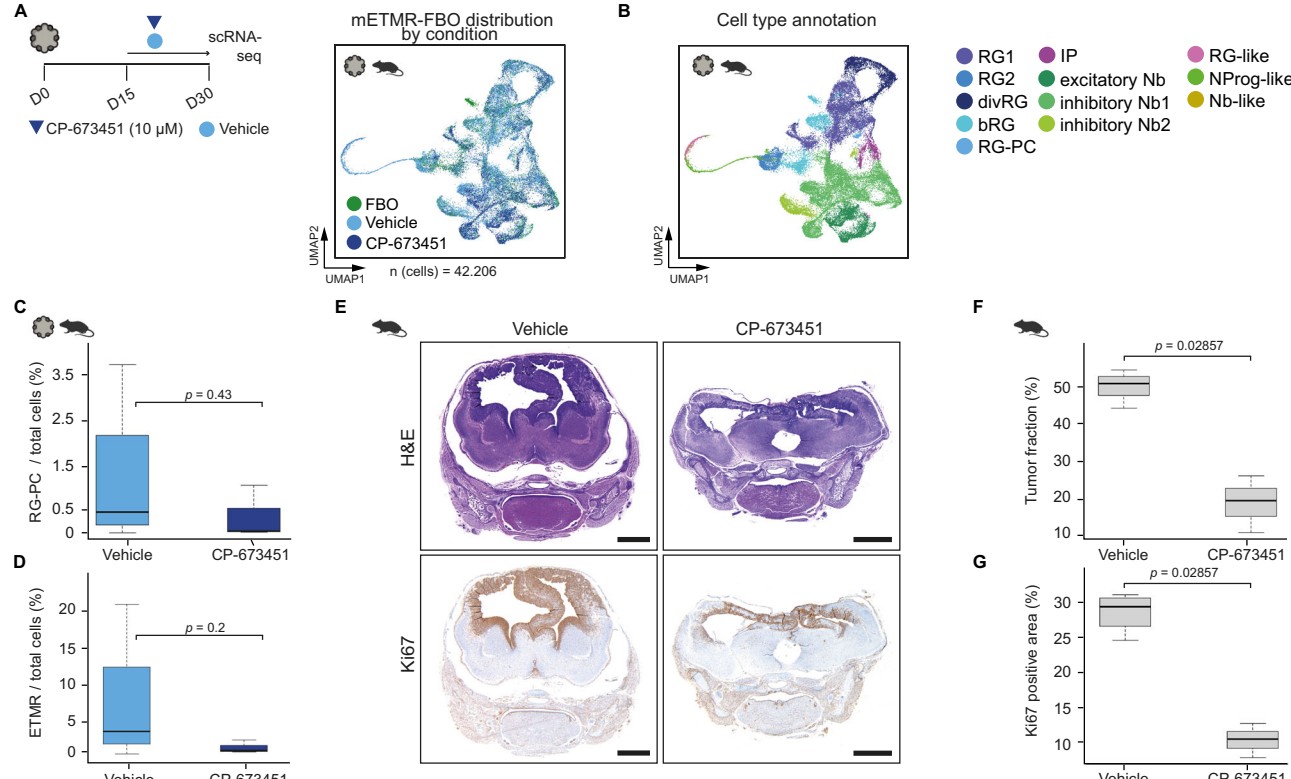

**Fig. 7 | PDGFR inhibition targets pericytes and decreases tumor cell population in FBO and in vivo. A** Left: mETMR-FBOs were treated with the PDGFR-inhibitor CP-673451 (10 µM) or vehicle at every medium exchange from day 15 (D15) to 30 (D30), followed by scRNA-seq at D30 (1 experiment). Right: UMAP of integrated dataset, colored by treatment - CP-673451 (*n* = 3), vehicle (*n* = 4) and untreated FBO controls (*n* = 2); *n* = biological replicates. **B** Cell type annotation of the mETMR-FBO dataset. ETMR subpopulations (RG-like, NProg-like and Nb-like) form distinct clusters from non-tumor FBO cells. **C, D** Boxplots showing the relative proportion (%) of PC cells (**C**) or ETMR cells (**D**) by condition in the dataset described in (**A**). Wilcoxon rank sum test, one-sided. Boxes represent the median (middle) and interquartile ranges (upper/lower hinges); whiskers show 1.5× IQR. **C** The PC population decreased from 0.44% (IQR 0.16–2.04) in vehicle-treated to 0.04% (IQR 0.03–0.52) in CP-673451-treated samples (*p* = 0.43). **D** The tumor population decreased from 3.87% (IQR 2.23–12.68) in vehicle-treated to 1.29% (IQR 1.19–2.02) in CP-673451-treated samples (*p* = 0.2). **E** Representative H&E and Ki67-stained histological images of ETMR-harboring murine brains at embryonic day E18.5, after 4 days of CP-673451 or vehicle treatment. Scale bar = 1 mm. **F, G** Quantification of tumor area (**F**) and ki67 positivity (**G**) in ETMR-bearing mice: vehicle-treated (*n* = 4) vs. CP-673451-treated (*n* = 3 animals); Wilcoxon rank-sum test, one sided. Boxes represent the median (middle) and interquartile ranges (upper/lower hinges); whiskers show 1.5× IQR. **F** Tumor fraction decreased in CP-673451-treated animals (mean = 18,78%; sd = 5,36) compared to vehicle-treated ones (mean = 48,09%; sd = 4,02). **G** Ki67+ areas decreased in CP-673451-treated animals (mean = 10,17%; sd = 1,72) compared to vehicle-treated ones (mean =2 28,61%; sd = 2,23). Source data of **C, D, F** and **G** are provided as a Source Data file. Sample type symbols in **A–C, E, F** created with BioRender.com.

(Fig. 7D). Similar trends were observed in hETMR-FBO treated with CP-673451 (Supplementary Fig. S21).

For in vivo validation, pregnant mice carrying ETMR-bearing embryos at embryonic day E14.5 were treated with CP-673451 or vehicle (see methods), and embryo brains were harvested for histological analysis at day E18.5 (Fig. 7E). This confirmed a significant reduction in tumor cell fraction (Fig. 7F) and Ki67 positive cell proportions (Fig. 7G) in treated embryos (*n* (animals) = 7), further supporting the effect of PDGFR inhibition on ETMR growth.

In summary, this study demonstrated that PDGFRβ⁺ pericytes contributed to ETMR aggressiveness by facilitating stem cell signaling in self-renewing tumor subpopulations, such as RG-like and NProg-like cells. In the TME of FBO, ETMR directed the differentiation of RG subpopulations into pericyte-like cells through a shift from WNT-high to PDGF-high signaling. PDGF signaling not only recruited PC into the TME but also enhanced survival in chemo-resistant RG-like tumor cells, thereby sustaining ETMR survival mechanisms through stem cell signaling and chemo-resistance (Fig. 8).

## Discussion

A developmental block has been previously described as characteristic of several pediatric brain tumor entities[8,56–58], where it was found to relate to distinct tumor properties, such as undifferentiated tumoral

cellular states presenting tumor initiation capacity[57,58], and/or to adverse prognosis[56]. In this study, we describe ETMR cellular states as three distinct tumor cell subpopulations (RG-like, NProg-like and NB-like) characterized by particular transcriptional, metabolic, and cell cycle programs. ETMR is predominantly composed of proliferating NProg-like cells, while RG-like cells are the ones driving chemotherapy resistance. These findings address the importance of the cells with self-renewing potential (both RG-like cells and NProg-like cells) for ETMR aggressiveness and, therefore, support the fundamental role of stem cell signaling in ETMR biology.

We explored the mechanism by which ETMR boosts stem cell signaling in its microenvironment. Notably, we found PC to co-localize with NSC-rich (SOX2⁺, NES⁺) tumor vascular areas and exchange paracrine signals through ECM interactions previously described to be enriched in niches of pluripotent cells and favoring the maintenance of self-renewal cell states [36,59–63]. Unfortunately, the lack of endothelial cells in our scRNA-seq datasets, likely due to our sample preparation approach, precluded further investigation into how L_R interacts with PCs may have influenced their gene expression and recruitment to the TME.

We unveil the differentiation of RG cells from the FBO into PC as one mechanism by which ETMR recruits PC to the tumor niche. PC are plastic cells[41,42], frequently associated with reprogramming

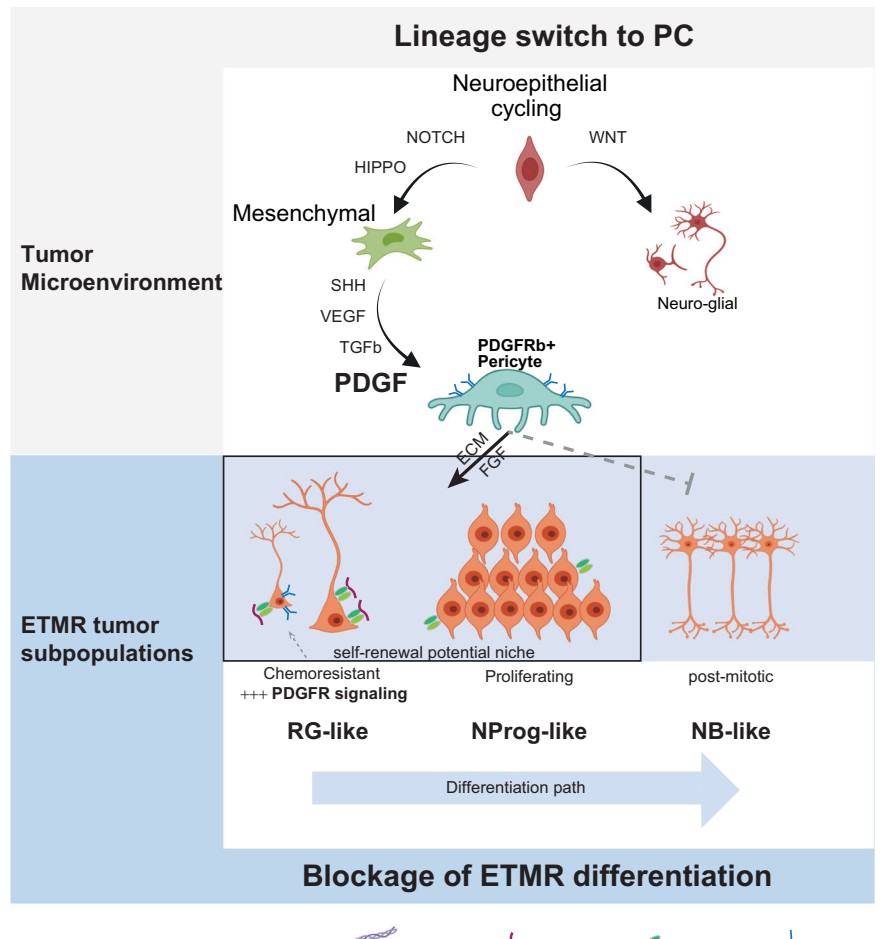

**Fig. 8 | Schematic of pericyte and PDGF signaling supporting stem cell maintenance and chemo-resistance in ETMR.** PDGFRβ+ pericytes and PDGF signaling support tumor survival through distinct mechanisms affecting both the tumor and the TME. In the TME: 1) ETMR promotes the transformation of neural stem cells (radial glial cells) into pericyte-like cells through a signaling cascade that shifts from WNT-high to PDGF-high expression; 2) PC communicates stem-cell signals through the ECM to maintain tumor sub-populations with self-renewal capacity (RG-like and NProg-like cells). In the tumor compartment: 1) Chemo-resistant RG-like tumor cells enhance PDGF signaling as a mechanism of survival advantage. Partially created with Biorender.com.

processes[16,64-66] and capable of arising from neuroepithelial cells of the neural crest[67,68], which are in line with our finding of RG-to-PC differentiation in our FBOs.

The discovery of an RG differentiation process leading to PC formation in the ETMR microenvironment was achieved by establishing a 3D culture model by integrating ETMR cells into FBO[39], which opened an opportunity to investigate interactions of ETMR with its TME. This platform led us to understand the trajectory of the RG-to-PC differentiation process induced by the tumor cells and reveal PDGFR signaling as a critical pathway for PC recruitment to the TME. Even in a chemically controlled microenvironment for neuronal differentiation, such as FBOs, tumor cells were still able to induce mesenchymal differentiation of local RG-to-PC, which highlights the significance of those cells in developing the tumor niche.

Notably, there is an interplay between PDGFR signaling and ETMR biology. Let-7 family miRNAs are suppressed in ETMR, leading to the upregulation of LIN28A, a molecular hallmark and a clinical marker of this tumor type[6]. Conversely, Let-7 miRNAs function as upstream regulators of PDGF signaling. There is in vitro evidence for the increased expression of PDGF and its receptor (PDGFR) in smooth muscle and endothelial cells upon suppression of let-7. Distinct cell lines treated with PDGF showed let-7 reduction, partially rescued by imatinib, a PDGF pathway inhibitor[53]. In the clinical setting, a second-generation PDGF inhibitor, dasatinib, induced

tumor response and long-term survival of a child with refractory ETMR[69].

We next investigated the potential mechanism of chemo-resistance in ETMR using chemotherapy on ETMR harboring murine embryos and forebrain organoids. Our results demonstrate that PDGFR signaling plays a crucial role to induce chemo-resistance, by favoring the survival of RG-like tumor cells to chemotherapy and enriching the TME with PC. Previously, PDGFRβ has been shown to be involved in chemo-resistance in glioblastoma[70] and fibrosarcomas[15], and in EMT in drug-resistant liver tumors[71], with antitumor effects of PDGFRβ inhibition[10,72].

Even though our study demonstrated the upregulation of PDGFR signaling in chemoresistant murine ETMR RG-like cells compared to untreated controls in silico, we did not generate chemoresistant cell lines to validate these findings in vitro or investigate the interplay between PDGFR signaling and ETMR genetic alterations. It remains unclear whether PDGFR upregulation is a stress-induced response or intrinsically related to the ETMR genetic background—representing a limitation of our study.

PDGFR inhibition in our study seemed to prevent ETMR-induced PC differentiation in FBO, although based on analyses of small cohorts. The absence of those cells disrupts the stem cell-like niche characteristic of ETMR, along with anti-proliferative effects on tumor cells in vivo. Overall, these findings represent a promising targeted

treatment strategy for ETMR. In addition, PC-targeting approaches could be employed to disrupt the blood-brain barrier, given PC role as a fundamental component of the neurovascular unit[41]. This disruption could facilitate drug delivery in the brain[73]. For example, inhibiting PDGFR, as described by Smyth et al.[74] may represent a promising strategy for this purpose.

In conclusion, our study showed that murine and human ETMR subpopulations recapitulate key stages of neuronal differentiation and possess distinct functional properties for tumor survival. PDGFRβ⁺ pericytes are crucial in maintaining a proliferative and stem-like niche in the ETMR microenvironment. Therefore, targeting PC via PDGFR inhibition represents a promising therapeutic approach for ETMR.

## Methods

### Ethical approval

This study complies with all relevant ethical regulations. All animal procedures were performed according to the guidelines provided by the local regulatory authorities (reference number 81-02.04.2018.A214; 81-02.04.2020.A474; 81-02.04.2021.A258; Government of NRW, Germany). Fresh tumor specimens were collected with signed informed consent of the respective patients´ parents or adult legal representatives per protocols approved by the Ethics Committee Münster (2017-261-f-S). Formalin-fixed paraffin-embedded (FFPE) tumor specimens were provided by Prof. Ulrich Schüller (University Medical Center Hamburg-Eppendorf) and Prof. Dr. Jens Schittenhelm (University of Tübingen, Germany) with a signed informed consent of the respective patients´ parents or adult legal representatives. As this study describes a very rare disease, we included all patients and did not select for age or sex/gender in advance. Sex/gender of patients was determined based on self-report.

### Genetically engineered mouse models

Genetically engineered *hGFAP-cre*, *SmoM2*$^{fl/fl}$, *Ctnnb1(ex3)*$^{fl/fl}$, and a *hGFAP-cre::Ctnnb1*$^{(ex3)fl/+}$*SmoM2*$^{fl/+}$ strains with a yFP reporter gene under the control of SmoM2 promoter have been previously generated and described[18]. All animals were housed and bred at the Central Animal Experimentation Facility of the University of Münster (Münster, Germany), in accordance with institutional and governmental regulations. Crossing *SmoM2*$^{fl/fl}$ with *Ctnnb1(ex3)*$^{fl/fl}$ mice, we obtained the heterozygous *SmoM2*$^{fl/+}$*Ctnnb1(ex3)*$^{fl/+}$ strain, which harbors no phenotype. By crossing *hGFAP-cre* mice with *SmoM2*$^{fl/fl}$*Ctnnb1(ex3)*$^{fl/fl}$ mice, we obtained *hGFAP-cre::Ctnnb1*$^{(ex3)fl/+}$*SmoM2*$^{fl/+}$ mice, in which Wnt and Shh pathways were constitutively activated, leading to ETMR development, as previously characterized histologically and at the gene expression level[18].

Genotyping was performed by PCR analysis using genomic DNA obtained from tail or cerebellum biopsies. Primers used to detect *Cre* transgene: Fw: 5´-TCCGGGCTGCCACGACCAA -3´and Rv: 5´-GGCGCGGCAACACCATTTT-3´. Primers used to detect *Ctnnb1*: Fw: 5´-CGTGGACAATGGCTACTCAA-3 and Rv: 5´-TGTCCAACTCCATCAGGTCA-3´. Primers used to detect *SmoM2*: Fw: 5´-GGAGCGGGAGAAATGGATATG -3´ and Rv: 5´-CGTGATCTGCAACTCCAGTC-3´.

Mice were housed in individually or in a maximum of 4 animals/cage in ventilated cages under specific pathogen-free (SPF) conditions with a 12-hour light/dark cycle (lights on at 7:00 a.m.). Ambient temperature was maintained at 20–24°C with a relative humidity of 45–65%. Animals had ad libitum access to food and water and were monitored daily for health and well-being.

As ETMR harboring mice die at P1 due to tumor extension[18], we obtained the forebrain of mice at either embryonic age of E16.5 or E18.5. Forebrains of mice at E16.5 were used for scRNA-seq, and those at E18.5 were used as a validation cohort for scRNA-seq and analysis of chemotherapy treatment (see below). To obtain the specimens, pregnant mice at E16.5 or E18.5 post-coitum (post-coital plug observation was considered as day 0.5), underwent cesarean section under

analgesia and sedation. Prior to the surgery, we monitored the pregnant mice daily for signs of distress (lethargy, weakness, pain). After the procedure, the mice were sacrificed by cervical dislocation. The embryos were decapitated for brain isolation.

### In vivo treatment experiments

All in vivo treatment experiments were performed in pregnant mice bearing *hGFAP-cre::Ctnnb1*$^{(ex3)fl/+}$*SmoM2*$^{fl/+}$ and *Ctnnb1*$^{(ex3)fl/+}$*SmoM2*$^{fl/+}$ embryos. The pregnant mice were weighed and monitored (twice) daily for distress signs (lethargy, weakness, pain) during the experiment using a defined score sheet. At E18.5, the animals underwent cesarean section for embryo extraction, embryo brain isolation, and genotyping for further studies. The pregnant mice were euthanized after embryo extraction with cervical dislocation under anesthesia. Embryos were immediately decapitated after cesarean. Treatment burden did not exceed the institutional endpoints, with no severe distress implied to the animals, in accordance with the guidelines defined by the animal authorities for this study. Embryos develop tumors normally between E13.0 and E15.0, but no maximum tumor size is defined for this model. In vivo *chemotherapy treatment*. One pregnant mouse was treated with etoposide (5 mg/kg, i.p; Gry), injected daily from day E14.5 to E16.5 and three cre+ embryos extracted through caesarian section were used for further experiments. Another pregnant animal was treated with corn oil in the same conditions and volumes, from which three cre−embryos were used as controls. In vivo *PDGFR-inhibitor treatment*. One pregnant mouse was treated with the PDGFR inhibitor CP−673451 (25 mg/kg/dose, i.p; #Cay19170-1) injected daily from day E14.5 to E17.5, from which 3 cre+ animals were used for experiments. Another pregnant mouse was treated with corn oil in the same conditions and volumes, from which four cre−embryos were used for further experiments.

### Human sample acquisition

Samples were diagnosed as ETMR by a reference pathology with standard clinical workup, 850k methylation classification, and molecular inversion probe CNV showing C19MC amplification (Supplementary Data 1–4). FFPE-embedded tumors with at least $4 \times 4$ mm in size were used for STs and immunohistochemistry analysis (Supplementary Data 2, 4).

### Cell lines

Murine ETMR cell lines (*hGFAP-cre::Ctnnb1(ex3)*$^{Fl/+}$*SmoM2*$^{Fl/+}$), used for co-culture and organoid experiments, were obtained from Prof. Dr. Ulrich Schüller (University Medical Center Hamburg-Eppendorf−UKE, Germany) and Prof. Dr. Kornelius Kerl (University Hospital Münster−UKM, Münster). Profesor Dr. Ulrich Schüller (University Medical Center Hamburg-Eppendorf−UKE, Germany) and Dr. Jennifer Chan also provided the human ETMR cell line (BT-183), which was used for ETMR-forebrain-organoids. We purchased the murine pericytes used for co-culture from ScienCell (mouse brain vascular pericytes, #M1200-57). Human induced pluripotent stem cells (hiPSC), used to generate forebrain organoids, were characterized in a previous study[75] and provided by Dr. Jan M. Bruder (Max Planck Institute for Molecular Biomedicine, Münster, Germany).

### Cell culture

To generate murine ETMR cell lines, fresh tumors of Cre+ animals were isolated from embryo brains by dissection and positive selection of the whole forebrain region (plus or minus midbrain) containing the tumor and exclusion of the hindbrain. The forebrain tissues were minced into pieces 1–2 mm³ in size, and enzymatically dissociated with papain solution (see single-cell dissociation for scRNA-sequencing, in methods). We cultured ETMR-neurospheres, both human and murine, in suspension six-well-plates or T25-flasks using DMEM/F12 medium (Gibco; #11330-057), supplemented with B27 (2%; Thermo Fisher;

#12587010), Pen/Strep (1%; Gibco; #15140-122), epidermal growth factor (EGF; 20 ng/ml; Peprotech; #315-09), basic fibroblast growth factor (bFGF; 20 ng/ml; Peprotech; #450-33) and Heparin (2 µg/ml; STEMCELL; #7980) as previously described[76]. We performed medium exchange every 3–4 days. When neurospheres reached a diameter >200 µm or presented with a dark core, we split them with Accutase (Sigma-Aldrich; #A6964) for 2 min at 37 °C followed by gentle pipetting. For culture maintenance, cells were reseeded at a density of 1:2 in the medium described above.

Murine pericytes were maintained in T75-standard cell culture flasks coated with Poly-L-Lysine (PLL, 2 µg/cm², ScienCell; #0413) in mouse brain vascular pericyte medium (ScienCell; #1231). Medium exchange was performed twice a week. When the cells reached 80–90% confluency, we split them with 0.05% Trypsin (Thermo Fisher; #25300054) and re-plated them at a density of 5000 cells/cm³.

The hiPS cells were cultured on Vitronectin (Thermo Fisher; #15134499) coated six-well-plates (Sarstaedt; #2068837) in mTesr plus medium (STEMCELL; #100-0276). Every other day, we exchanged the medium. We split iPSCs with Accutase (Sigma-Aldrich; #A6964) when reaching 80–90% confluence and re-plated them at a density of 1:10 to 1:20 in mTesr Plus medium supplemented with ROCK inhibitor Y-27632 (10 µM, tebu-bio; #331-10373-2) for the first day post-split.

All cell lines were periodically tested to be negative for mycoplasma by PCR analysis.

## ETMR-PC co-culture experiment

On experimental day 0 (D0), PC were dissociated with 0.05% Trypsin (Thermo Fisher; #25300054) and 40,000 cells were seeded in a PLL-coated well of a 24-well plate with 500 µl of PC medium to allow cells to adhere to the plate. On D1, the supernatant was discarded, and the wells were washed once with 500 µl PBS. For co-culture, ETMR neurospheres were dissociated with Accutase (Sigma-Aldrich; #A6964) and 100,000 cells were seeded in suspension in PC-pre-seeded wells filled with 500 µl of ETMR medium. For monoculture, we seeded the same number of ETMR cells from the same cell suspension in PLL-coated wells filled up with 500 µl ETMR medium, as per co-culture. After 72 h and for all the conditions, cells adhering and in suspension were harvested, resuspended in PBS/1% BSA, FACS-sorted for living cells with 7-aminoactinomycin D (7AAD; Thermo Fisher; #00-6993-50) with a BD FACSAria II Cellsorter (BD Biosciences), and used as input for scRNA-seq analysis.

## Forebrain organoids

For our automated forebrain organoid culture, we used a protocol published previously[39]. In brief, we manually split hiPS cells with Accutase (Sigma-Aldrich; #A6964) after reaching 90% confluence. All following steps were performed using an automated liquid handling system (Beckman Coulter) with an attached incubator (Thermo Fisher). On day 0, 10,000 hiPSC cells were seeded per well in an ultra-low-attachment U-bottom-plate (Corning; #CLS7007-24EA) using the liquid handling system in mTesr Plus medium (STEMCELL; #100-0276) supplemented with ROCK inhibitor Y-27632 (10 µM, tebu-bio; #331-10373-2) and 0.4% w/v polyvinylalcohol (PVA, Sigma-Aldrich; #363170).

Media changes were performed on D1 after seeding, D4, and then every other day with cortical organoid medium containing DMEM/F12 (Gibco, #11330-057), Knock-Out Serum Replacement (20%; Gibco; #10828028), Pen/Strep (1%, Gibco; #15140-122), Glutamax (1%; Gibco; #35050061), non-essential-amino-acids (1%; Gibco; #11140-050) and 2-mercaptoethanol (100 µM; Gibco; #11528926). We supplemented the medium until D6 with dorsomorphin (5 µM; Enzo Life Science; #ENZ-CHM141-0010) and SB-431542 (10 µM;; Biomol; #Cay13031). From D6-D24, the medium was supplemented by EGF and FGF2 (both 20 ng/ml, Peprotech; #315-09 and #450-33) and from D24-D45 by NT3 and BDNF (both 20 ng/ml, Peprotech; #450-03 and #450-02).

## Generation of ETMR-forebrain organoids

For the generation of our ETMR-forebrain organoids (ETMR-FBO), we split hiPSCs and either murine (GEMM-derived cell line) or human (BT-183 cell line) ETMR cells with Accutase. After cell counting, we seeded a total of 10,000 cells per well, hiPSCs and ETMR cells in a ratio of 10/20:1 (20:1 for murine ETMR cells and 10:1 for human ETMR cells), in mTesr Plus medium (STEMCELL; #100-0276) supplemented with 0.4% w/v PVA (Sigma-Aldrich; #363170) and ROCK inhibitor Y-27632 (10 µM, tebu-bio; #331-10373-2) in ultra-low-attachment U-bottom-plates (Corning; #CLS7007-24EA). After seeding ETMR-forebrain organoids, we followed the standard forebrain organoid differentiation protocol. In the first experiment, we tried co-aggregating hiPSCs with mETMR and murine PC. This approach failed, as no PC gene expression was retrieved by scRNA-seq analysis, and samples with or without PC did not differ transcriptionally. Those samples have been, therefore, integrated into the scRNA-seq to characterize the murine mETMR.

## ETMR-forebrain organoids chemo-treatment

For the chemotherapy drug testing, we treated mETMR-FBOs and hETMR-FBOs for 96 h starting on either day 20 of differentiation or day 24 with one of the following: DMSO 0.5% (PanReac AppliChem; #A3670), 10 µM Carboplatin (Gry), or 1 µM Etoposide (Sigma-Aldrich; #E1383), based on previously described drug toxicity profile[76]. We replenished the treatment medium after 48 h and exchanged it after 96 h for medium without drugs. We applied three different intervals following treatment to perform organoid dissociation for scRNA-seq: short, intermediate and late. For short- and late-interval regimes, treatment was given from organoid culture day 24 to day 28, and dissociation took place on day 30 or day 48, respectively. In the intermediate interval, organoids were treated from day 20 to day 24 and dissociated on day 30.

## ETMR-forebrain organoids PDGFR inhibitor treatment

To explore the inhibition of radial glia cells' differentiation to pericytes, we treated the ETMR-FBO from D15 to D30 with the selective PDGFR-inhibitor CP-673451 (10 µM, Sellekchem; #Cay19170-1) in standard forebrain medium. Medium containing the drug in the appropriate concentration was exchanged every second day.

## Single-cell RNA sequencing of murine, human, 2D and organoid samples

**Sample preparation.** A fresh biopsy of human ETMR and fresh murine forebrains of Cre+ and Cre- ETMR embryos at E16.5 and E18.5 gestational age were isolated for single-cell dissociation. Samples were minced into pieces of 1–2 mm³ size with scalpels and digested for 30 min at 37 °C in papain solution consisting of papain (CellSystems; #LK003178) in pre-warmed DMEM/F12 (Gibco, #11330-057) with 32.2 µg/ml DNAse (Roche; #11284932001). After the digestion phase, samples were triturated, passed through a 40 µm cell strainer (Corning; #32110302), and red blood cells were depleted using ACK lysis buffer (Thermo Fisher; #A1049201), according to the manufacturer´s instructions. Fresh murine and cryopreserved human cell suspensions were used as input for scRNA-seq, after prior sorting for vital cells using 7-Aminoactinomycin D (7-AAD, Thermo Fisher; #00-6993-50) staining in flow cytometry with a BD FACSAria II Cellsorter (BD Biosciences). Forebrain organoids were pooled as eight single organoids per sample for scRNA-seq analysis. We adapted our protocol for organoid single-cell dissociation on D30 or D48 by placing the intact organoids in papain solution for 60 min instead of 30 min. We supported the dissociation in the first 30 min with shaking every 5 min and for the second 30 min with regular gentle pipetting using wide-bore tips. After the digestion phase, cell suspensions were passed through a 70 µm cell strainer (Miltenyi; #130-095-823) and were freshly used as input for scRNA-seq without previous FACS sorting and a vitality of >95% each.

**Sample cohorts**. Samples from murine forebrain, human tumor, ETMR-PC co-culture, FBO, and ETMR-FBO were used as input for scRNA-seq (Supplementary Data 1). For the murine dataset analysis, we used mFB harboring ($n = 2$) and non-harboring ETMR ($n = 2$) samples. We cell-dissociated one human tumor for the human dataset and merged the dataset with another published human ETMR[8]. For 2D culture, each well of a 24-well plate of monoculture or co-culture was processed as one individual dataset. For all organoid analyses, each sample was composed of 8 whole-forebrain organoids, cell-dissociated as a pool for further scRNA-seq processing.

**Library preparation**. Single-cell suspensions were processed using Chromium Single Cell 3' Gel Bead Kit v3.1 (PN-1000121, 10x Genomics) and a Chromium Next GEM Chip G Single Cell Kit (PN-1000127, 10x Genomics) according to the manufacturer's protocol. In brief, cell suspensions were loaded onto a Chromium single-cell controller (10x Genomics) to generate gel beads in emulsion (GEMs). The number of loaded cells varied by sample. Captured cells were lysed, and the released RNA was barcoded through reverse transcription in individual GEMs. Following GEM-RT, barcoded cDNAs were cleaned up with Dynabeads, amplified, and size selected through SPRIselect beads cleanup. The Library Bead Kit (PN-1000157) and Chromium i7 Multiplex Kit (PN-120262) were used for generating single-indexed single-cell libraries for Illumina sequencing. Double-indexed libraries were generated using Library Construction Kit (PN-1000190) and Dual Index Kit TT Set A (PN- 3000431). Quality, purity, size, and concentrations of cDNA and libraries were determined by Tapestation 2000 (Agilent Technologies, Inc.). Libraries were sequenced using the NextSeq 500 sequencing platform (high-throughput Kit, 75 Cycles v2 Chemie) for the scRNA-seq of primary tumors and ETMR-PC co-culture. The organoid scRNA-seq series and the STs datasets were processed in NextSeq 2000 instrument (high-throughput kit, 100 cycles) at the Genomics Core Facility (University Hospital Münster, Germany).

### Fixed RNA sequencing
Formalin-fixed, paraffin-embedded human ETMR tumor samples were prepared for single-nuclei fixed-RNA profiling using the Single Cell Gene Expression Flex protocol CG000632_rev.B (10x Genomics). Briefly, after deparaffinization and rehydration, the tissue sections were dissociated using a heated gentleMACS Octo dissociator for FFPE tissue (130-096-427; Miltenyi Biotec) with freshly prepared Dissociation Enzyme Mix containing Liberase TH (5401151001; Millipore Sigma). After sample filtration, single-nuclei suspensions were counted using the Invitrogen Countess 3 Automated Cell Counter (16842556; Thermo Fisher Scientific). To generate fixed RNA gene expression libraries, we used the Chromium Next GEM Single Cell Fixed RNA Sample Preparation Kit, 16 rxn (PN-1000414). Single-nuclei suspensions were mixed with human transcriptome probes and hybridized for 20 h at 42 °C and then further processed according to protocol CG000527_RevE for multiplexed libraries (10X Genomics). One million cells per sample were used for probe hybridization. GEMs were prepared with a targeted recovery of 80.000 cells in each 16-plex library.

### Bioinformatic QC, Normalization, Clustering, and Differential Gene Expression of Single-Cell Data
The distinct scRNAseq datasets were analyzed using the actualized and concurrent R version[77] and Bioconductor packages and summarized in the Supplementary Data 1. The package Seurat[78] was used for quality control, filtering, dimensionality reduction, clustering, and differential expression analysis. The reads were aligned to the mouse reference genome (mm10, v1.2.0) with the CellRanger pipeline, v2.0.2[79]. Next, we merged the datasets using Seurat function *MergeSeurat*. All count data was normalized and integrated with the related Seurat functions. The

sample composition for all single-cell/nuclei integrations described in the manuscript is presented in the Supplementary Data 8. The batch correction was performed for all integrations on a sample level and using the anchor-based integration method from Seurat.

For the different ETMR organoid sample datasets, the raw scRNA data were aligned to a combined human and murine reference genome GRCh38 and mm10-2020-A with CellRanger v6.0.2[79], and feature matrices were calculated with CellRanger's count function using default values. The count data was then normalized, integration anchors were calculated, and the samples were integrated for each dataset separately using Seurat functions and default parameters with Seurat v4.0.5[80] and R v4.0.5[81] for each dataset. Thresholds for basic quality filtering were defined for each dataset (Supplementary Data 8). Furthermore, outlier cells showing a very high nCount_RNA number relative to other cells were classified as doublets and filtered out accordingly. For both the human ETMR and the ETMR organoid datasets, clusters with ambiguous or split annotations were subclustered with basic Seurat functions until a sufficient resolution was achieved. Differential markers between chosen conditions or clusters were identified using Seurat's findMarkers with a two-sided MAST statistical test. The integration of the chemotherapy-treated mETMR-FBO and hEMTR-FBO was performed as previously described by ref. 82. Further marker plots, annotations, and cell cycle plots were created with Seurat functions, ggplot2 (https://ggplot2.tidyverse.org), and iSEE[83]; dot plots for chosen gene sets were visualized with Seurat's wrapper function. Additionally, heatmaps were created with the R package pheatmap (https://CRAN.R-project.org/package=pheatmap). For the murine ETMR data, the correlation analysis was performed using the Pearson correlation and based on two input gene sets: i) all differential genes per chosen cell type, and ii) ETMR subpopulations gene signatures (Supplementary Data 5–7). R's cor.test function and Bonferroni correction for multiple testing were used to calculate $p$ values for chosen cell types. A pseudotime trajectory plot was created with Monocle3[84], using radial glia type 1[40] as a start cluster, and default values otherwise. All pseudotime values were exported at the cell level, and visualized further with R. Gene modules were identified with Monocle3's find_gene_modules function and a resolution parameter of 0.001. Finally, CellPhoneDB v2.1.5[32] and InterCellar[33] were used to predict and visualize receptor-ligand interaction pairs.

### Bulk RNA sequencing
Microarray bulk RNA data from Kool et al.[7] (GEO accession number GSE122077, link https://www.ncbi.nlm.nih.gov/geo/query/acc.cgi?acc=GSE122077) were downloaded and analyzed with the R packages affy[85] and limma[86]. The data were rma-normalized, and further processed in R. Mean signature scores were calculated for chosen sets of genes from the expression data, and subsequently subjected to a linear transformation to the interval. Heatmaps were created for these normalized signature scores with the R package pheatmap.

### STs
FFPE tissue blocks of ETMR tumors from distinct patients were submitted to neuropathology review and analyzed for RNA quality. Three tissue sections of distinct tumors with up to $6.5 \times 6.5$ mm$^2$ were selected for placement on the mRNA capture areas of a Visium Gene Expression slide for FFPE (2000233, 10x Genomics), fixed in methanol and H&E stained according to the Visium Gene expression for FFPE user guide (CG000409 Rev A, 10x Genomics). Brightfield histology images were taken using a 4x objective on a Nikon Eclipse Ni (5254 × 5396 pixels). Raw images were stitched together using NIS-Elements (Nikon) and exported as.tiff files with high-resolution settings. Libraries were prepared following the Visium Spatial Gene expression for FFPE user guide (G000407 Rev A, 10x Genomics) and library size and concentration were measured using a Tapestation

2000 (Agilent Technologies, Inc). The library was sequenced on a NextSeq 2000 instrument (Illumina high-throughput kit, 100 cycles), with a sequencing depth of ~25,000 pair-reads per spot in the following read protocol: read 1: 28 cycles; i7 index read: 10 cycles; i5 index read: 10 cycles; and read 2: 50 cycles.

## Visium raw data processing

Alignment and general preprocessing of ST data were conducted with the 10x Genomics Space Ranger software suite v1.3.0 against the human reference genome GRCh38 2020-A and default parameters. Filtered feature matrices were integrated with Seurat v4.0.5 and R v4.0.5, and spatial marker plots and differential gene lists were created with the related Seurat functions for relevant clusters and marker genes.

## Histology and immunostaining

We used sections of FFPE samples from patients, murine embryo brains and forebrain organoids for histological analysis and immunostaining. Hematoxylin and eosin (H&E) stainings were performed according to standard protocols[87]. For immunohistochemical stainings, the ultraView, OptiView, or iView DAB Detection Kit (all Roche Diagnostics) was used on a Ventana Benchmark xt System (Roche Diagnostics) according to the manufacturer's instructions. A list of the used antibodies can be found in Supplementary Data 9. According to the manufacturer's instructions, fluorescence in situ hybridization (FISH) was performed using the C19MC/TPM4 FISH Probe Kit (CT-PAC03, Cytotest). For image acquisition, any adjustment was applied to the entire image, and no threshold manipulation was applied.

## Quantitative immunohistochemistry

Four paraffin-embedded human tumor tissues were utilized for quantitative analysis of CD13+ pericyte density in ETMR tissue. After deparaffinization, tissue slides were steam heated for 16 minutes at 95EC in EDTA buffers (pH 8) for antigen retrieval, followed by pretreatment with peroxidase inhibitor, and the subsequent immunostaining with a 24 min incubation period of 50 µl CD13 antibody counterstained with a biotinylated secondary antibody (OptiView DAB IHC Detection Kit, Ventana Medical Systems, Inc.) and hematoxylin, as per standard protocols. We analyzed 325 unique fields of view from the four histological sections using Vectra® 3.0 multispectral imaging system and the image analysis software Phenochart™ and InForm® (PerkinElmer) classifier, from Akoya Biosciences, which were used for automated image acquisition, tissue segmentation, cell phenotyping, and biomarker quantification. First, whole slide scans (×10 magnification) were performed using the brightfield acquisition mode. For PC density scoring, 60–90 images (per sample) of a size of 0.669 mm = 0.500 mm (field of view) were manually chosen across the whole section of one sample for a re-scan at ×20 magnification. All areas enriched in tumor tissue were positively selected. Areas only containing paraffin or enriched in artifacts were negatively selected. Subsequently, we performed automated tissue segmentation (hypercellular, hypocellular or artifact regions) and cell segmentation (nucleus, cytoplasm), followed by active phenotyping of CD13+ cells with the learning InForm ® phenotyping function. We trained the system multiple times until achieving an identity confidence of over 70% for DAPI or CD13 staining, which were considered for calculating the frequency of all cells (DAB/Hematoxylin+) per mm² as well as pericytes (CD13+) cells per mm². All tissue samples were analyzed blinded to the patient's clinical background. Statistical analysis was performed with a one-sided Wilcoxon rank sum test with continuity correction.

## Immunohistochemistry of in vivo PDGFR treatment

Measurement of tumor size in H&E stainings and of Ki67 signal in DAB stainings from treated and untreated animals was performed with ImageJ (v 1.48a). To evaluate tumor size, H&E overview pictures of frontal sections from comparable levels were selected. Total brain areas and cell-dense tumor areas were selected by hand and measured using the ImageJ *Measure* tool. For Ki67 evaluation, sections from the same level were selected and stained with the automated Ventana system on the same run to ensure comparability of signals. Overview pictures were taken, and ImageJ DAB color deconvolution was applied. The resulting images (Color 2) were converted into 8-bit format. The area corresponding to a positive Ki67 signal was measured using a black/white mask with thresholds (0, 200).

## Forebrain organoids, whole-mount immunostaining and tissue clearing

For histology and immunostaining, we fixed the organoids in 4% PFA for 15 min at room temperature. We washed the samples three times with PBS and stored them until further processing for up to half a year at 4 °C in PBS.

We followed a previously published protocol for whole-mount immunostaining and tissue clearing[88] with minor adaptions. In brief, we first incubated the samples for 6 days with the primary antibodies (1:500 anti-Map2: ab32454, abcam; 1:500 anti-GFP: ab13970, abcam) and 6 days with the secondary antibody (Alexa Fluor 488 donkey anti-rabbit, Alexa 647 goat anti-chicken, 1:1000, Thermo Fisher) and DAPI (0.5 µg/ml; Sigma-Aldrich; #2040116) diluted in blocking and permeabilization solution 6% BSA (Thermo Fisher; #15260037), 0.5% Triton X-100 (Roth; # 3051.1), 0.1% (w/v) sodium azide (Sigma-Aldrich; #71289) in PBS at 37 °C. The antibody solutions were replaced by fresh solutions every second day. Between primary and secondary antibody incubation steps and after secondary antibody incubation, samples were washed five times with 0.1% Triton X-100 (Roth; # 3051.1) diluted in PBS. Second, we dehydrated the samples with a methanol (Roth; #4627.5) series (two times 30 min each, 25%, 50%, 75%, 90%, and 100%). Followed by transferring the samples to a solvent-resistant cyclo-olefin-coated 96-well plate ("Screenstar", Greiner Bio-One; #655866). Lastly, the samples were incubated for 120 min in 1:1 v/v Methanol/BABB (benzyl aldehyde (Sigma-Aldrich; #305197) and benzyl benzoate (Sigma-Aldrich; #B6630), 1:1 v/v), with an exchange of fresh solution after 60 min. Samples were then kept in BABB at 4 °C until analysis.

## Imaging of organoid samples

Confocal imaging was performed on an Operetta high content imager (Perkin Elmer) at ×10 magnification. Exposure times were evaluated for each experiment and applied equally to all samples from one plate. Confocal optical slices were obtained with a minimum Z-distance of 30 µm in order to prevent double-counting of segmented structures. Each plate contained negative controls with samples that did not contain ETMR cells and samples that contained ETMR cells but no primary antibody to assess the background level of staining. We analyzed the images in Columbus version 2.6.0 (Perkin Elmer) using a maximum projection of all confocal planes after automatic flatfield correction. Tumor cells expressed YFP, which was quenched using BABB clearing. YFP signal was specifically detected using an anti-GFP/YFP primary antibody, amplified with an Alexa Fluor 647 secondary antibody (see details above). Alexa Fluor fluorophores continue to fluoresce under BABB clearing conditions.

## Multiplex immunofluorescence staining

**Samples and sample preparation.** Tissue sections from one ETMR FFPE tissue block with an area of ~6 × 6 mm² were cut, mounted onto a super frost slide, and incubated at 42 °C for 3 hours. Deparaffinization was performed in a descending alcohol series followed by heat-induced antigen retrieval using Tris-EDTA buffer at pH 9 and 40 min incubation in a pressure cooker. Next, MACSima Running Buffer (Miltenyi Biotec, #130-121-565) was used for blocking for 60 minutes at

room temperature before probing with primary antibodies diluted in MACSima Running Buffer overnight at 4 °C. This was followed by washing of the slides with MACSima Running Buffer, pre-staining of nuclei using DAPI Staining Solution (Miltenyi Biotec, component of MACSima Stain Support Kit, #130-127-574) to incubate the slides with the diluted solution for ten minutes in the dark at room temperature, and a final washing step. Finally, a ~3 × 2 mm$^2$ ROI (neuropathologist assignment) was gated for imaging, and further antibody staining was performed using the MACSima Imaging System according to its user manual. A list of all antibodies used in the experiment can be found in the Supplementary Data 9.

**Processing and analysis.** The acquired images were stitched and pre-processed using the image analysis software MACS iQ View (Miltenyi Biotec) before further processing. No processing or averaging was performed, which enhances the resolution of the image. All post-processing adjustments were applied to the entire image, and no non-linear adjustments/gamma changes were made. Cell segmentation was performed using the ImageJ (NIH) plugin StarDist[85], and the segmentation mask imported to MACS iQ View. The same software was used for all cell gating and cell-cell distance analysis: pericyte regions were defined as the areas limited by PDGFRB$^+$ signal. Cell populations were gated based on their mean signal intensity for the channels of interest. Pericyte distances were calculated for every gated cell population and data exported to R version 4.4 to generate a histogram plot.

### Statistics and reproducibility
Methods used for statistical hypothesis testing and exact *n* numbers are directly stated in the figure legends. The significance level was set to 0.05. Where applicable, *p* value corrections were also displayed in figure legends. Boxplots were generated using the function boxplot of the package graphics R version 4.4.1, which show median (center line), first and third quartile (bounds) and minima/maxima (whiskers) of all measurements. Boxplot settings (middle = median; lower hinge = 25% quantile; upper hinge = 75% quantile; upper/lower whisker = largest/smallest observation less/greater than or equal to upper/lower hinge ±1.5 = IQR).

### Reporting summary
Further information on research design is available in the Nature Portfolio Reporting Summary linked to this article.

## Data availability
Raw single-cell RNA-seq and spatial transcriptomic generated in this manuscript have been deposited in the Gene Expression Onmnibus database (GEO) under the accession number GSE224478. Raw Fixed RNA-seq data have been deposited under the accession number GSE254819. Single-cell RNA-seq data from one human ETMR sample were obtained from Jessa et al. (2019) and are available in the European Genome-Phenome Archive under accession number EGAS00001003368 under request. Bulk-RNA sequencing data were obtained from Lambo et al. (2019) and is publicly available in GEO database under the accession number GSE122077. Murine cell lines can be readily available upon request to the corresponding author. ETMR-forebrain organoids can be obtained upon request from the corresponding author. Source data are provided with this paper.

## Code availability
This study used established codes, which are available in the following links: 1) Seurat: https://satijalab.org/seurat/ or https://cloud.r-project.org/web/packages/Seurat/index.html; 2) CellPhoneDB V2: https://github.com/Teichlab/cellphonedb; 3) InterCellar: https://www.bioconductor.org/packages/release/bioc/html/InterCellar.html; 4) CellRanger: https://support.10xgenomics.com/single-cell-gene expression/software/pipelines/latest/installation; 5) CON-ICSmat: https://github.com/diazlab/CONICS.

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

## Acknowledgements

We thank all members of the K.K. lab for discussions, technical support, and critical reading of the manuscript. We thank Daniela Jeising (Department of Pediatric Hematology and Oncology, University Children's Hospital Münster, Münster, Germany) for processing scRNA-seq and supervising the processing of ST samples. We thank Annegret Rosemann and Elisabeth Jung (Department of Pediatric Hematology and Oncology, University Children's Hospital Münster, 48149 Münster, Germany) for the FACS sorting of samples. We thank the Core Facility Genomik (Medical Faculty of Muenster) for the partnership in data sequencing. Parts of bioinformatic processing for this study were performed on the HPC cluster PALMA II of the University of Münster, subsidized by the Deutsche Forschungsgemeinschaft (INST 211/667-1). N.C.R. received funding from "Medizinerkolleg Münster" (21-0007). K.K. is supported by funds from the Wilhelm Sander Stiftung (2023.023.1), the Deutsche Kinderkrebsstiftung (DKS 2023.06) and the Deutsche Forschungsgemeinschaft (DFG KE2004/4-1). The U.S. is generously funded by the Fördergemeinschaft Kinderkrebszentrum Hamburg. Figures were partly created with BioRender.com released under a Creative Commons Attribution-NonCommercial-NoDerivs 4.0 International license.

## Author contributions

F.W.F., N.C.R. conducted research, performed experiments, data analysis and wrote the manuscript. D.M., M.I. and C.W. conducted bioinformatic coding. V.M. performed experiments and held a supervising role. C.G., L.A., M.R., C.T., M.S., I.B., N.M., M.G., M.H., W.H. and P.H.P. gave direct assistance in experiments and revised the manuscript. W.H., A.K.B., A.B., F.L.S., B.M., D.H., O.S., A.C., M.E., M.S., J.S. and U.S. contributed to the patient cohort and histopathological assignment. R.R., S.S., J.V., M.D., S.T.B., I.L., C.R. and M.F.F. revised the manuscript. TKA contributed to the data analysis, gave direct assistance in conducting experiments, had a supervision role and revised the manuscript. CW conducted and supervised the bioinformatic coding and revision of the manuscript. JMB contributed to the organoid study design, data analysis, revision of the manuscript and held a supervising role. KK designed and directed the study, provided funding, supported data interpretation and revised the manuscript.

## Funding

## Competing interests

The authors declare no competing Interests.

## Additional information

Flavia W. de Faria[1,19], Nicole C. Riedel[1,19], Daniel Münter [1], Marta Interlandi[1,2], Carolin Göbel [3,4], Lea Altendorf[3,4], Mathis Richter [5], Viktoria Melcher[1], Christian Thomas [6], Rajanya Roy[1], Melanie Schoof [3,4], Ivan Bedzhov [7], Natalia Moreno[1], Monika Graf[1], Marc Hotfilder[1], Dörthe Holdhof [3,4], Wolfgang Hartmann [8,9], Ann-Katrin Bruns[10], Angela Brentrup[10], Friederike Liesche-Starnecker[11], Bruno Maerkl[11], Sarah Sandmann[2], Julian Varghese [2], Martin Dugas [2,12], Pedro H. Pinto[13], Sebastian T. Balbach [1], I-Na Lu [1], Claudia Rossig[1], Oliver Soehnlein [5], Aysegül Canak[14], Martin Ebinger [15], Martin Schuhmann[15], Jens Schittenhelm [16], Michael F. Frühwald [17], Ulrich Schüller [3,4], Thomas K. Albert [1], Carolin Walter[1,2,20], Jan M. Bruder[18,20] & Kornelius Kerl [1,20] ✉

[1]Department of Pediatric Hematology and Oncology, University Hospital Münster, Münster, Germany. [2]Institute of Medical Informatics, Westphalian Wilhelms University Münster, Münster, Germany. [3]Department of Pediatric Hematology and Oncology, University Medical Center Hamburg-Eppendorf, Hamburg, Germany. [4]Research Institute Children's Cancer Center Hamburg, Hamburg, Germany. [5]Institute for Experimental Pathology, Center for Molecular Biology of Inflammation, University of Münster, Münster, Germany. [6]Institute of Neuropathology, University Hospital Münster, Münster, Germany. [7]Embryonic Self-Organization Research Group, Max Planck Institute for Molecular Biomedicine, Münster, Germany. [8]Division of Translational Pathology, Gerhard-Domagk Institute of Pathology, University Hospital Münster, Münster, Germany. [9]West German Cancer Center (WTZ), Network Partner Site, University Hospital Münster, Münster, Germany. [10]Department of Neurosurgery, University Hospital Münster, Münster, Germany. [11]Department of Neuropathology, Pathology, Medical Faculty, University of Augsburg, Augsburg, Germany. [12]Institute of Medical Informatics, Heidelberg University Hospital, Heidelberg, Germany. [13]Department of Pathology, Children's Hospital of Brasilia Jose de Alencar, Brasilia, Brazil. [14]Department of Hematology and Oncology, Children's University Hospital Tübingen, and German Cancer Consortium (DKTK) Tübingen, Tübingen, Germany. [15]Department of Neurosurgery, section of Pediatric Neurosurgery, University Hospital Tübingen, and German Cancer Consortium (DKTK) Tübingen, Tübingen, Germany. [16]Department of Neuropathology, University Medical Hospital, Eberhard Karls University Tübingen, Tübingen, Germany. [17]Swabian Children's Cancer Center, Pediatric and Adolescent Medicine, University Center Augsburg, Augsburg, Germany. [18]Department for Cell and Developmental Biology, Max Planck Institute for molecular Biomedicine, Münster, Germany. [19]These authors contributed equally: Flavia W. de Faria, Nicole C. Riedel. [20]These authors jointly supervised this work: Carolin Walter, Jan M Bruder, Kornelius Kerl. ✉e-mail: kornelius.kerl@ukmuenster.de

