## [Peer Review File · Nature Communications]

ETMR stem-like state and chemo-resistance are supported by perivascular cells at single-cell resolution

Corresponding Author: Professor Kornelius Kerl

Version 0:

Reviewer comments:

Reviewer #1

(Remarks to the Author)

This manuscript by Faria et al reports on experiments to define the cellular states and signaling pathways at the single cell level in EMTR tumours, which represent a rare but lethal brain tumour in children. Through single cell RNA sequencing of human tumors and genetically engineered mouse tumours, they show that EMTR contains 3 main cell states that are also found in the murine forebrain including RG-like, NProg-like, and Nb-like cells. Whereas Nb cells were dominant in healthy murine embryonic forebrain, NProg-like cells were most abundant in the tumours suggesting a block in differentiation to maintain tumor cells in a stem like proliferative state. This block in the neural differentiation is similar to what is seen in many other types of brain tumors including glioblastoma. Importantly in these tumours there was also differentiation of radial glia into pericytes, which further increased angiogenesis. These pericytes had increased PDGF signaling, and by ligand receptor pair analysis they showed that pericytes drove many cellular interactions in the tumor microenvironment. Of clinical relevance, they found that chemotherapy (etoposide) resistance was correlated with increased PDGFR signaling in radial glia like cells in these tumors, and inhibition of PDGFR signaling reduced the number of pericytes and tumor cells therefore lowering tumor development both in an organoid and mouse model.

This paper addresses an area of clinically unmet need, and presents novel data regarding neurodevelopmental lineage block which fits with pattern seen in other brain tumors. The strengths of the paper include the use of multiple models and patient samples, and the single cell resolution achieved with transcriptomic analysis. The paper is well written and the figures are presented clearly. I only have some minor issues that should be addressed prior to publication:

1. PDGFR signalling is frequently driven by genomic amplification in many brain tumours; it would be useful to report on whether this is the case here.
2. Is there any correlation between increased PDGFR signaling and poorer survival in patients with EMTR?
3. For the last figure, it seems unclear as to what time point has been checked for the reduction in tumor fraction in mice treated with a PDGFR inhibitor. Does this translate to improved survival in these mice?
4. Pericytes contribute to the blood brain barrier. It would be worth commenting on issues surrounding targeted drug treatment and blood brain barrier penetration in this tumor type, with reference to recent literature discussing these issues.
5. Figure 6 legend has P values stated with a comma instead of a decimal point make sure needs correction.

Reviewer #2

(Remarks to the Author)

I believe the most noteworthy result of this study, that may have implications beyond ETMR, is 1) the conversion of a radial glial-like cells into pericytes, and 2) that these newly formed pericytes promote tumor stemness and thus cancer cell cycling and treatment resistance.

What is not clear to me from my readings of this paper are:

- 1) are the RG-like cells that are changing states to PCs cancer cells?
- 2) is there evidence that the pericytes derived from these RG like cells also cancerous - sharing the same CNAs.

Assuming 1 and 2 above are both true - it would be very important to show that this occurs in the tumor in situ. The overall signature of an RG and pericyte is very different - using single cell resolution transcriptomics or multiprotein labeling, it

would be more convincing if the authors were able to identify cancer RGs and PCs in proximity. My reading of the results and supplementary data is that that has not been shown. Similarly, showing the proposed signalling between the 2 would also be important in situ.

Reviewer #3

(Remarks to the Author)

In “ETMR stem-like state and chemoresistance are supported by perivascular cells at single-cell resolution” (NCOMMS-24-14528” de Faria and Ridel et al. investigate cellular communication in the tumor microenvironment (TME) of Embryonal Tumor with Multilayered Rosettes (ETMR) using scRNAseq of human and murine ETMR, in vitro cultures, and 3D organoid models. The study describes parallels between developmental trajectories that are present in the mouse forebrain and those in mouse and human ETMR, argues for a stall in a proliferative differentiation state in ETMR, and suggests that ETMR cells induce radial glia (RG) cells to differentiate into pericytes which in turn promote tumor stemness and angiogenesis. Targeting the PDGF signaling axis that supports this program and which is upregulated in chemotherapy-resistant ETMR cells may block RG differentiation into pericytes and tumor growth.

This is a largely a well-done and comprehensive study that uses both mouse models and human samples to investigate an aggressive pediatric brain tumor that is very difficult to study given how rare it is. There is a large amount of data including 22 supplementary figures which is largely clear to this reviewer until Figure 4 (with some suggestions for the part on Fig 1-3 below), in part due to the need for more clarity in the writing especially given that effects are described on both non-tumor (RG and PC cells) and tumor cells (RG-like cells). With some more effort in the description of Figures 4-6, this should be a strong study.

However, in the current version, a combination of the wording in the text and the figure panels gets rather difficult to understand. Given that the experimental results of the transition from RG to PC in lines 252-261 appears to be very important to the paper, it should be rewritten to be less confusing and more clearly stated. Is the argument that RG differentiate into PC (not neural progenitor cells into PC as listed in the subtitle of the section in line 253 – confusing given that there is a lot on NP tumor cells in this paper)? Is ‘interfere’ (line 255) the correct word? How do the GO terms of RG cells show an association with PC function? Where do the RG-PC and RG-ANGIO categories arise from? What are they? Is this identified in this data or a well-known state of RG differentiation? Is ‘process’ the correct work in line 261 (is that process referring to an effect of tumor cells on nontumor cell differentiation)? Not everyone is an expert in this niche of neuroscience and more clarity in the text would make this less obscure. How certain are the authors of the presence of this transitional population?

Figure 5 is equally challenging. The treatment nomenclature (D24_D30, D20_D30, and D24_D48) is confusing as is the description of the results (line 297 to 316) in Fig 5D and E - is there a simpler way to integrate the data rather than single gene violin plots? It would help to simply get to the point that RG-like tumor cells are likely the resistant population, and work on the text to be more clear. Lines 328 – 331 are unclear. What is ‘apoptosis recovery-related genes’ – I can assume that this is gene program linked to chemoresistance – but it would be easier for the reader if you describe it as “a specific set of genes involved in the cellular processes that enable cells to recover and survive after an apoptotic signal has been initiated but not completed”. Is the writing here crystal clear to all of the co-authors? Please make this easier to understand. Since you are seeing PDGFR signaling as important in both the non-tumor (found to be linked to the RG to PC differentiation that occurs in the presence of tumor cells) and tumor cells, please point that out to signal this at key points in the transitions in the writing about the later figures. “Based on the previous observations” (line 343) is not sufficient. It is worth spending time to be more specific to describe the results and the logic. “hETMR-FBO treated PDGFR inhibitor dataset” = “hETMR-FBO treated with CP-673451”

Additional clarity in the text will be useful – for instance, please discriminate more clearly which systems are human v. mouse (2D co-culture, FB organoid model) when first introducing them; the addition of the mouse and human (and model) symbols in the figures was very useful.

It might be worth considering that the enrichment of the Nprog-like cells in ETMR is not due to a stall, but rather to enhanced fitness of those cells in the tumor.

The authors appropriately use the Visium data in a circumspect manner; it would be helpful if they quantify the difference in proliferation (Ki67) between Nprog-like areas v. RG-like and Nb-like areas. It is not clear from the inferred ligand-receptor interaction analysis that the outgoing PC interactions are mainly to immature (RG, NP) rather than also to mature (NB) cells so it is possibly worth rephrasing that. Also, it would help to more clearly state the reason for the focus on PC; presumably the EC L-R interactions seen in the human data were not followed because of the inability to detect EC in mouse? Please list in the text the quantification of the increase in NP cells shown in Fig. S11F and statistical analysis (stats needed elsewhere too such as Figure 6C and 6D). It will be helpful to know if this increase in NP cells following co-culture with PC cells has been repeated and is a robust and reproducible result.

Can the authors address the difference in CD13 PC between hyper and hypocellular parts of ETMR; does the detected difference simply reflect a higher density of vessels in hypercellular regions, or are the vessels in the hypercellular regions more likely to have pericytes than the vessels in hypocellular areas of the tumor?

It is not clear what effect co-culture with FBO derived from hiPSC has on the identify of the human BT-183 and mouse GEMM-derived EMTR tumor cells which they are co-cultured with. How do the fractions of RG, NP, and NB-like cells change from before and after co-culture? Maybe I missed it.

Changes would help to the abstract and to the discussion to clearly list the identities of the three distinct ETMR tumor cell populations (RG-like, NProg-like, and Nb-like). A schematic summarizing the author's model of tumor and vascular populations and their crosstalk would be useful. While it is helpful to have much of the supporting data in the supplementary figures, it might help to move some additional key aspects of the study to the main figures (perhaps a new Figure 2 that contains elements of the supplemental figures 4-8).

Reviewer #4

(Remarks to the Author)

The paper "ETMR stem-like state and chemoreistance are supported by perivascular cells at single-cell resolution" by de Faria et al., is an important paper that captures the cellular heterogeneity of ETMRs at exquisite detail, using a combination of primary tumor samples, organoid models and xenografts. The paper highlights the crucial role of the TME in ETMRs and challenges the current paradigm which often posits ETMRs as being immunologically cold. While this is a novel finding, the study suffers from biologically relevant mechanistic insights that would help further therapeutic approaches, as detailed below -

Major comments:

- a. The existing literature around ETMRs show conclusively that the oncogenic drivers of ETMRs are an upregulation of C19MC amplification, coupled with DICER1 mutations that lead to a dysregulation of miRNA's. The current study, shows a new role of oncogenesis in ETMRs – one that involves the WNT-PDGF signaling cascade. However, the novel pathway alluded in this study, fails to connect to the existing literature on C19MC induced ETMR oncogenesis. It would be helpful if the authors can address whether such a link exists and how the link can be manipulated with PDGFR inhibition. This is a crucial question, as much of the clinical decision making, in terms of ETMR recurrence stems from an evaluation of the C19MC status. If a patient fails conventional therapy, as they often do, would selective PDGFR inhibition play a role in stemming cancer growth? Also, which of the 3 sub-populations of ETMRs house the C19MC cells? This is also relevant as much of the current strategies are aimed at eliminating these oncogenic cells. Without a mechanistic link to C19MC, the paper lacks potential therapeutic strategies that are clinically relevant.
- b. The current study makes use of several bioinformatics strategies to deduce link to VEGF, TGF, PDGF, apoptosis, etc signaling. However, they lack any biochemical or cell biological validation. Over-expression of genes does not imply activation of these pathways. The latter can be validated using phospho-specific antibodies (on Westerns, or IHC) that specifically addresses whether the pathway is active in ETMR cell-lines.
- c. The authors mention a large component of stem cell signaling in ETMRs between pericytes and tumor cells, but then point to pericytes driving a neurogenic-like proliferative state. As neuronal progenitors are not stem-like, these two statements seem to be at odds.
- d. The authors demonstrate a reduction of PC cells when treated with PDGFR inhibitor CP-673451. However, I question if a reduction from 0.44% of the TME to 0.04% of the TME serves as a biologically significant change.
- e. Furthermore, we would not expect pericytes in organoid models, but treating the organoid models with PDGFR inhibitors leads to reduction in tumors cells and proliferative Ki-67 positive tumors cells, reinforcing the author's claims that PDGFR is important in ETMR growth but weakens their own claims that pericytes play a specific role in PDGFR signaling.
- f. Authors claim EMT in Figure 4E but do not look into canonical EMT transcription factors such as ZEB, TWIST and SNAIL.
- g. I would like to see the enrichment of the PC signatures on the 10X spatial data, to further validate the presence of pericytes.
- h. I would also like see some form of spatial organizational analysis to investigate cell-cell signaling with the spatial information integrated. I would also ask if spatial distribution of pericytes is significant, or if a random spatially distributed pericyte population would lead to similar significance.

Minor comments

- a. The authors use the data presented in Suppl figure 8 to validate the transcriptomic heterogeneity of 3 primary ETMRs. However, there is heterogeneity among the three ETMRs to make any meaningful conclusions. For example, the ETMR signature correlates strongly with Lin28A in T2, but not with Ki-67, and even less with NProg-like signature or Nb-like signature. In T1, there are multiple populations with RG-like, Nprog-like and Nb-like signatures, which does not really correspond to the ETMR signature.
- b. In Figure 2E, the authors state that PC-ETMR signatures are "mainly" to Rg-like and N-prog-like population. Judging from the pie-chart, in the absence of any provided numbers, one would conclude that the interactions are fairly similar between PC and three ETMR sub-populations.
- c. In Figure 2G, the contributions of NB-like cells interactome with PC cells and EC cells are seemingly ignored, with the focus shifting primarily to RG-like and Nprog-like cells. How do the authors determine significance of interactions, is it purely subjective?
- d. Supplementary figure 11F – Based on the figure and the accompanying legend, with a p value of >0.05, and with the large error bars in mono-culture, one would conclusively state that the pericytes do not change the proportions of NP-Prog like cells. However, the authors conclude that pericytes drive tumor cells to a neurogenic-like proliferative state. On what basis is this conclusion made?
- e. In Supplemental figure 11 D – Stmn4 is replaced by Stmn2 between the mono-culture vs. co-culture conditions, precluding one to make a conclusion about the differences in expression between the stemness genes in different sub-

populations.

f. In Suppl figure 13 – the authors conclude that pericytes co-localize with hyper-cellular regions of the ETMR. Do they see a similar co-location with endothelial cells – In other words, is the co-localization of pericytes in the hyper-cellular areas due to the enrichment of blood vessels in the hyper-cellular zones?

g. In Figure 5D – There is a decrease in RG-like signatures in Day 30 untreated vs Day 48 untreated controls. Do the authors have any explanations of why the RG-like population fluctuate between the two controls, and how this can potentially affect their interpretation of the data.

h. In Figure 5D – what happens to the Nb-like cells following treatment with Etoposide/Cisplatin?

i. Using tSNE could confound how transcriptionally distinct or not the clusters are. I would like to see UMAP reduction in Figure 1 to be more up to date.

j. It is unclear what samples contribute to the pericyte populations in Figure 1B, as cluster numbers are missing from the tSNE

k. Colors are too similar and hard to differentiate (Figure 2F, Figure 6B)

l. Technical documentation could be more detailed as to batch corrections on which combinations of datasets

m. I wonder if integrating the FFPE samples alongside the fresh frozen human samples would provide additional insights into the heterogeneity of ETMRs, or would the technologies be too distinct for integration.

n. Please italicize “in vitro” and “in vivo” throughout the manuscript.

Reviewer #5

(Remarks to the Author)

I co-reviewed this manuscript with one of the reviewers who provided the listed reports. This is part of the Nature Communications initiative to facilitate training in peer review and to provide appropriate recognition for Early Career Researchers who co-review manuscripts

Reviewer #6

(Remarks to the Author)

Version 1:

Reviewer comments:

Reviewer #1

(Remarks to the Author)

The authors have fully addressed all of my comments. I congratulate the authors for this important work and endorse publication.

Reviewer #2

(Remarks to the Author)

My comments have been adequately addressed.

Reviewer #3

(Remarks to the Author)

The authors have addressed my comments by substantially improving the clarity of the figures and the text – particularly regarding terminology and experimental setups. The additional quantification is useful, and the schematic is another valuable addition to help readers understand the rather detailed aspects of this manuscript. I understand the authors' point against relocating the detailed supplementary data. I think it would help to add a sentence regarding the rationale for not pursuing endothelial cell L-R interactions in EMTR.

Reviewer #5

(Remarks to the Author)

Reviewer 4 comments –

The paper has been modified based on the reviewers suggestions and the addition of clarifying data and statements makes the paper easier to read.

1. However, the paper lacks mechanistic insight, which I feel is the main drawback of the paper. It fails to draw a link to the current dogma of ETMRs being driven by C19MC/Dicer1 mutations. While I understand the technical limitations of not being able to detect miRNA in single cell technologies, Lin28 can potentially be used as a surrogate marker. Jessa et al (Nature

genetics 2019) used TTYH1 as a surrogate marker. Furthermore, the addition of the C19MC pre-mRNA as a pseudogene has been demonstrated to be a robust readout of C19MC activity. (also same reference)

2. None of the assays that have been used in this study are functional. The authors have access to ETMR cell lines (BT183). Analyzing the functional status of the PDGF pathway and/or analyzing the survival of these cells in face of PDGFR inhibition would lend more credence to the main assertions of this study.

3. Thank you for making these changes.

4. Thank you for making these changes.

5. Thank you for making these changes.

6. Thank you for making these changes.

7. This analysis suggests that indeed the PC signature is tumor-related, but then it is somewhat in disagreement with the previous result, where it states that TME (non-malignant) PCs are what drive signaling. I would like some clarification on if this PC signaling is tumor or TME related.

8. While the pericyte signature is enriched within the tumor mass on a spatial level, it still isn't clear if this cell-cell signaling is occurring. I would like to see some plot of L-R signaling as predicted by single cell to validate that cells are indeed expressing these LR genes, and that they are closely spatially organized supplement the multiplex IHC.

9 - 21 - Thank you for making these changes.

Reviewer #6

(Remarks to the Author)

Point-by-point responses to referee comments

We deeply appreciate the time and effort that all reviewers and editors have invested to evaluate our work and we thank the editorial board for its overall positive evaluation on our study “**ETMR stem-like state and chemo-resistance are supported by perivascular cells at single-cell resolution**”. The reviewers’ relevant questions and constructive suggestions have significantly improved the quality of our manuscript. We carefully addressed all the comments, providing considerable additional value to the readership of *Nature Communications*.

We made extensive changes in the manuscript text to increase the clarity of concepts, hypothesis and interpretation of experimental results, as a general request from distinct reviewers. In addition, we included new validation experiments, which were incorporated in the body of the manuscript. The changes implemented in this revision resulted in changes in the composition of main and supplementary figures. We summarize below the new data added to the revised manuscript:

New Data	Reviewers point	Figure
In situ relationship between pericytes and RG-like, NProg-like and Nb-like tumor subpopulations	Reviewer 2 point 2 Reviewer 3 point 11 Reviewer 4 point 8	Fig. 3A-C Supplementary Fig. 11G
Quantification of differences in proliferation (Ki67) between Nprog-like areas vs RG-like and Nb-like areas in spatial transcriptomics	Reviewer 3 point 7	Supplementary Fig. S11F
Schematic summarizing the author’s model of tumor and vascular populations and their crosstalk	Reviewer 3 point 14	Fig. 8
Enrichment of the PC signatures on the 10X spatial data	Reviewer 4 point 7	Supplementary Fig. S11G

ETMR-FBO threatment analysis	chemotherapy improved	Reviewer 3 point 2 Reviewer 4 point	Fig. 6A-B Supplementary Fig. 18I-J
--------------------------	--	---------------------------------------

Please find below the point-by-point answer to the reviewers’s comments.

Reviewer 1:

“This manuscript by Faria et al reports on experiments to define the cellular states and signaling pathways at the single cell level in EMTR tumours, which represent a rare but lethal brain tumour in children. Through single cell RNA sequencing of human tumors and genetically engineered mouse tumours, they show that EMTR contains 3 main cell states that are also found in the murine forebrain including RG-like, NProg-like, and Nb-like cells. Whereas Nb cells were dominant in healthy murine embryonic forebrain, NProg-like cells were most abundant in the tumours suggesting a block in differentiation to maintain tumor cells in a stem like proliferative state. This block in the neural differentiation is similar to what is seen in many other types of brain tumors including glioblastoma. Importantly in these tumours there was also differentiation of radial glia into pericytes, which further increased angiogenesis. These pericytes had increased PDGF signaling, and by ligand receptor pair analysis they showed that pericytes drove many cellular interactions in the tumor microenvironment. Of clinical relevance, they found that chemotherapy (etoposide) resistance was correlated with increased PDGFR signaling in radial glia like cells in these tumors, and inhibition of PDGFR signaling reduced the number of pericytes and tumor cells therefore lowering tumor development both in an organoid and mouse model.

This paper addresses an area of clinically unmet need, and presents novel data regarding neurodevelopmental lineage block which fits with pattern seen in other brain tumors. The strengths of the paper include the use of multiple models and patient samples, and the single cell resolution achieved with transcriptomic analysis. The paper is well written and the figures are presented clearly. I only have some minor issues that should be addressed prior to publication:”

Response: We thank the reviewer for this positive feedback and address the specific points below.

- 1) “PDGFR signalling is frequently driven by genomic amplification in many brain tumours; it would be useful to report on whether this is the case here.”

Response: We appreciate the reviewer insightful comment. According to the literature there is no evidence for a genomic amplification of PDGFR in ETMR ^{2,6,7}. However, there are evidence in the literature for the interplay between PDGFR signalling and ETMR biology. We included this information in the sections “**PDGFR signaling is harnessed by RG-like tumor cells to evade chemotherapy-induced cell death in vitro and in vivo**” (page 15, lines 325-327) and “**Discussion**”, where a more detailed explanation was given (page 17, lines 390-397), as follows: “Notably, there is an interplay between PDGFR signaling and ETMR biology. Let-7 family miRNAs is suppressed in ETMR, leading to the upregulation of LIN28A, a molecular hallmark and a clinical marker of this tumor type ⁶. Conversely, Let-7 miRNAs function as upstream regulators of PDGF signaling. There is in vitro evidence for the increased expression of PDGF and its receptor (PDGFR) in smooth muscle and endothelial cells upon suppression of let-7. Distinct cell lines treated with PDGF showed let-7 reduction, partially rescued by imatinib, a PDGF pathway inhibitor ⁵⁶. In the clinical setting, a second-generation PDGF inhibitor, dasatinib, induced tumor response and long-term survival of a child with refractory ETMR ⁷².”

- 2) “Is there any correlation between increased PDGFR signaling and poorer survival in patients with EMTR?”

Response: To address this point, we analysed a publicly available small cohort of ETMR patients (n = 28, GEO accession number GSE122077) for which gene expression and clinical data including overall survival was provided. We present below the results relative to the reviewer’s question. Since the cohort is small and the result did not reach statistical significance, we refrained from any assumptions related to survival and we did not include these results in our revised manuscript.

- 3) “For the last figure, it seems unclear as to what time point has been checked for the reduction in tumor fraction in mice treated with a PDGFR inhibitor. Does this translate to improved survival in these mice?”

Response: We apologize for the lack of clarity in providing the time-point of the experiment. To clarify this information, we added the following sentence to the manuscript (page 16, lines 348-352): “For in vivo validation, pregnant mice carrying ETMR-bearing embryos at embryonic day E14.5 were treated with CP-673451 or vehicle (see methods) and embryo brains were harvested for histological analysis at day E18.5 (Fig. 7E).”

Due to the murine model inducing tumor development during the embryonic stage, which leads to a lethal phenotype within the first days postnatally, ethical constraints prevented us from assessing long-term survival of the animals.

- 4) “Pericytes contribute to the blood brain barrier. It would be worth commenting on issues surrounding targeted drug treatment and blood brain barrier penetration in this tumor type, with reference to recent literature discussing these issues.”

Response: We appreciate the reviewer’s contribution to this point. Indeed, therapeutic strategies targeting the blood-brain barrier (BBB) via pericyte and PDGFR signalling inhibition have been explored. To address this in the manuscript, we included the following discussion on page 18, lines 410-413: “In addition, PC-targeting approaches could be employed to disrupt the blood-brain barrier, given PC role as a fundamental component of the neurovascular unit⁴³. This disruption could facilitate drug delivery in the brain⁷⁶. For example, inhibiting PDGFR, as described by Smyth et al. (2022), may represent a promising strategy for this purpose⁷⁷.” Furthermore, in the context of ETMR, we added the following comment on page 18, lines 397-398: “In the clinical setting, a second-generation PDGF inhibitor, dasatinib, induced tumor response and long-term survival of a child with refractory ETMR⁷².”

- 5) “Figure 6 legend has P values stated with a comma instead of a decimal point make sure needs correction.”

Response: We revised the manuscript thoroughly as to consistently present all p-values starting with a decimal point.

Reviewer 2:

“I believe the most noteworthy result of this study, that may have implications beyond ETMR, is 1) the conversion of a radial glial-like cells into pericytes, and 2) that these newly formed pericytes promote tumor stemness and thus cancer cell cycling and treatment resistance.”

We appreciate the reviewer’s observation about the implications of our study beyond ETMR and we hope our study to bring insights into mechanisms of therapy resistance associated to cell interactions in the microenvironment involving cancer stem cell phenotypes of other entities.

- 1) “Are the RG-like cells that are changing states to PCs cancer cells?”
- 2) Is there evidence that the pericytes derived from these RG like cells also cancerous - sharing the same CNAs.

Assuming 1 and 2 above are both true - it would be very important to show that this occurs in the tumor in situ. The overall signature of an RG and pericyte is very different - using single cell resolution transcriptomics or multiprotein labeling, it would be more convincing if the authors were able to identify cancer RGs and PCs in proximity. My reading of the results an supplementary data is that that has not been shown. Similarly, showing the proposed signalling between the 2 would also be important in situ.”

Response: We apologize for the lack of clarity in some parts of the manuscript, leading to misinterpretation of the data. We did not hypothesize that tumor cells (RG-like) could transform into pericytes. Instead, we explored two related hypotheses: first, the potential role of pericytes in enhancing stemness within ETMR, and second, the capacity of ETMR to influence RG cells from the tumor microenvironment (TME) to differentiate into pericytes, thereby recruiting them to its microenvironment.

As this is a critical point in our manuscript, we implemented significant overall changes the manuscript text to increase clarity. Of note, we point here the main changes:

- In the introduction section: we state now clearly in the end of the introduction our working hypothesis. On page 4, lines 49-54, we added: “In this study, we explored ETMR cell heterogeneity on single-cell level to understand mechanisms of chemoresistance and explore two related hypothesis: first, the potential role of

pericytes in enhancing stemness within ETMR and, therefore, chemoresistance, and second, the capacity of ETMR to influence RG cells from the tumor microenvironment (TME) to differentiate into pericytes, thereby recruiting them to its microenvironment.”

- To familiarize the reader to the name of the tumor subpopulations, we substituted the term “tumor subpopulations” to “RG-like”, “NProg-like” and /or “NB-like” in the abstract (page 2, line 15)
- We have reordered several paragraphs in the first result section, titled “Single cell transcriptomics of murine and human ETMRs identified three distinct subpopulations of malignant cells with a predominance of self-renewing states” (page 4, line 56), to enhance clarity. Specifically, the composition of Fig. 1 was adjusted to simplify the description of RG-like, NProg-like and Nb-like tumor cell subpopulations. In addition, we now present murine and human data separately to improve navigation through the results. The murine ETMR- related findings are shown in Fig. 1B-D and Supplementary Figs. S1-S4, while human ETMR data are presented in Fig. 1E-F and Supplementary Figs. S5-S6. Corresponding paragraphs in the text have been rearranged to align with this revised sequence of presented results.
- In the result section (page 11, lines 245-246), we revised the title “**ETMR interferes with the cell fate determination in FBO by favoring the differentiation of neuronal progenitors into pericytes through a WNT-VEGF-PDGFR pathway**” to “**ETMR induces a lineage transition from non-malignant RG to PC via a WNT-VEGF-PDGFR pathway**”. This modification aims to clarify that the lineage transition process was related to non-malignant RG cells.
- In the same section above described, we clarified the non-malignant origin of the cells undergoing lineage transition and assigned specific names to distinguish them from both malignant cells and pericytes. This clarification is stated on page 11, lines 247-249: “We next explored interactions between tumor and FBO cells, identifying non-malignant RG subpopulations that differed between control FBO and mETMR-FBO. In the mETMR-FBO, two unique RG-FBO clusters, termed RG-PC and RG-angio”.
- We addressed the reviewer’s request regarding the spatial relationship between tumor subpopulations and pericytes by conducting multiplex immunofluorescence (Fig. 3A-C) and further spatial analysis (Supplementary Fig. 11). Spatial transcriptomic data from FFPE tumor sections revealed an enrichment of pericyte gene expression predominantly in RG-like-enriched regions (Supplementary Fig. S11G). Furthermore, protein expression of cell type markers highlighted a clear proximity between self-renewing tumor cells (RG-like, NProg-like) and perivascular regions enriched with pericytes (Fig. 3A-C)

Reviewer 3:

“In “ETMR stem-like state and chemoresistance are supported by perivascular cells at single-cell resolution” (NCOMMS-24-14528” de Faria and Ridel et al. investigate cellular communication in the tumor microenvironment (TME) of Embryonal Tumor with Multilayered Rosettes (ETMR) using scRNAseq of human and murine ETMR, in vitro cultures, and 3D organoid models. The study describes parallels between developmental trajectories that are present in the mouse forebrain and those in mouse and human ETMR, argues for a stall in a proliferative differentiation state in ETMR, and suggests that ETMR cells induce radial glia (RG) cells to differentiate into pericytes which in turn promote tumor stemness and angiogenesis. Targeting the PDGF signaling axis that supports this program and which is upregulated in chemotherapy-resistant ETMR cells may block RG differentiation into pericytes and tumor growth.

This is a largely a well-done and comprehensive study that uses both mouse models and human samples to investigate an aggressive pediatric brain tumor that is very difficult to study given how rare it is. There is a large amount of data including 22 supplementary figures which is largely clear to this reviewer until Figure 4 (with some suggestions for the part on Fig 1-3 below), in part due to the need for more clarity in the writing especially given that effects are described on both non-tumor (RG and PC cells) and tumor cells (RG-like cells). With some more effort in the description of Figures 4-6, this should be a strong study.”

Response: We apologize for the lack of clarity in the manuscript and we thank the reviewer for the positive assessment of the relevance of our study. We also appreciate the reviewer’s identification of several opportunities of improvement, facilitating our understanding of the critical points requiring clarification. In response, we have addressed the reviewer’s comments throughout the manuscript, leading to substantial revisions in the writing style, while maintaining the integrity of the overall content.

- 1) “However, in the current version, a combination of the wording in the text and the figure panels gets rather difficult to understand. Given that the experimental results of the transition from RG to PC in lines 252-261 appears to be very important to the paper, it should be rewritten to be less confusing and more clearly stated. Is the argument that RG differentiate into PC (not neural progenitor cells into PC as listed in the subtitle of the section in line 253 – confusing given that there is a lot on NP tumor cells in this paper)? Is ‘interfere’ (line 255) the correct word? How do the GO terms of RG cells show an association with PC function? Where do the RG-PC and RG-ANGIO categories arise from? What are they? Is this identified in this data or a well-known state of RG differentiation? Is ‘process’ the correct work in line 261 (is that process referring to an effect of tumor cells on nontumor cell differentiation)? Not everyone is an expert in this niche of neuroscience and more clarity in the text would make this less obscure. How certain are the authors of the presence of this transitional population?”

Response: We are happy to respond below to the specific points raised above by the reviewer regarding the section presented now in pages 11-13, lines 245-283:

We rephrased most of the section “**ETMR-forebrain organoids recapitulate ETMR intratumoral heterogeneity defined by scRNA-seq**” (pages 10-11, lines 216-244), to bring more clarity to the forebrain organoid model experimental set up, hopefully supporting the understanding of further results related to the reviewer’s questions. The overall content of the section, however, was not changed.

- To address the questions raised regarding the content in the previous lines 253 and 255, we changed the section’s title “**ETMR interferes with the cell fate determination in FBO by favoring the differentiation of neuronal progenitors into pericytes (...)**” to “**ETMR induces a lineage transition from non-malignant RG to PC via a WNT-VEGF-PDGFR pathway**”, now presented in page 11, lines 245-246. This change was made to specify the cell type undergoing lineage transition and to better highlight the role of ETMR cells in this process.
- To distinguish between non-malignant and malignant RG cells in this section, we explicitly refer to non-malignant cell populations as “RG cells of the FBO”. In addition, we designated the unique non-malignant RG cells of the FBO as “RG-PC” and “RG-angio” to clearly differentiate them from malignant RG-like cells. We have revised several lines in this section to enhance clarity on this distinction, as illustrated on page

12, lines 247-249: “We next explored interactions between tumor and FBO cells, identifying non-malignant RG subpopulations that differed between control FBO and mETMR-FBO. In the mETMR-FBO, two unique RG-FBO clusters, termed RG-PC and RG-angio, emerged (...).” Those terms were used consistently throughout subsequent sections and also in the figures (e.g, Fig. 5A-C; Fig. 7).

- To address the clarity regarding the gene ontology (GO) similarities between PC and RG-PC, we have described their association with ECM organization terms and cited the relevant figures that show the GO terms of both cell types. This is detailed on page 12, lines 249-251: “(...) two unique RG-FBO clusters, termed RG-PC and RG-angio, emerged based on their association with either ECM organization – resembling PC GO terms (RG-PC), or blood vessel development (RG-angio) (Supplementary Fig. S7B, S14E, S15A-B).”
- We also sought to clarify that RG-PC and RG-angio were not expected in the FBO based on previous single-cell RNA sequencing studies of brain organoids described in the literature (e.g., Kelava et al. (42)). In our own FBOs, we saw a marked expression in those cell types in the ETMR co-aggregated organoids, in comparison to FBO controls (Fig. 5A-B, Supplementary Fig. S15C). These cell types, therefore, likely emerged as a result of cell interactions with ETMR cells. This point is described on page 12, lines 252-255: “These novel RG subpopulations, while not expected in FBOs⁴², were present in tumor-co-aggregated FBOs (Fig. 5A-B, Supplementary Fig. S15C). Notably, the emergence of PC-like gene expression in RG-PC cells suggests a tumor-induced lineage shift toward a pericyte-related state.”
- We employed distinct approaches to confidently assign this transitioning cell population by: i) verifying the expression of pericyte-specific gene from primary tumors within the FBO dataset; ii) analysing the activation of signalling pathways consistent with a pericyte-like phenotype; iii) defining a lineage trajectory that mechanistically explains the phenomenon. All this points are addressed in page 12, lines 252-275. Notably, we identified RG-PC in a second independent cohort of mETMR-FBO, which we used for PDGFR inhibition experiments in vitro (Fig. 7B, Supplementary Fig. 20E, see below).

2) “Figure 5 is equally challenging. The treatment nomenclature (D24_D30, D20_D30, and D24_D48) is confusing as is the description of the results (line 297 to 316) in Fig 5D and E - is there a simpler way to integrate the data rather than single gene violin plots?”

Response: In respond to the above request, we have integrated the mETMR-FBO and hETMR-FBO datasets, refined the description and nomenclature of the distinct treatment intervals, and reanalysed the data using expression scores derived from RG-like, NProg-like and Nb-like gene signatures instead of individual genes. Additionally, we have updated the main figure (Fig. 6, see below) to focus exclusively on the results for etoposide treatment, while moving the remaining results, which are primarily confirmatory, to the supplement (Supplementary Fig. S18). The overall figure composition and accompanying text (page 13, lines 286-304) have been significantly revised and conclusions streamlined for clarity. We now describe the main findings as follows (lines 294-300) “In the short interval, we observed a proportional increase in Nb-like gene expression, likely reflecting the expected lower sensitivity of non-cycling cells to chemotherapy. At the intermediate interval, during which surviving cells had the opportunity to emerge post-chemotoxicity, there was a notable enrichment in the expression of RG-like cells, alongside a modest increase in NProg-like cells and decrease of Nb-like cells in comparison to untreated D30 organoid controls (Fig. 6B). This suggests that RG-like cells are the best suited to survive chemotherapy-induced cell injury and, therefore, associated with chemotherapy resistance.”

We hope these modifications address to the reviewer’s concerns satisfactorily.

- 3) “It would help to simply get to the point that RG-like tumor cells are likely the resistant population, and work on the text to be more clear. Lines 328 – 331 are unclear. What is ‘apoptosis recovery-related genes’ – I can assume that this is gene program linked to chemoresistance – but it would be easier for the reader if you describe it as “a specific set of genes involved in the cellular processes that enable cells to recover and survive after an apoptotic signal has been initiated but not completed”. Is the writing here crystal clear to all of the co-authors? Please make this easier to understand.”

Response: We thank the reviewer for the suggestion to make our statements more concise and precise. In response, we have revised several lines to provide a clearer explanation of the association between RG-like cells and a chemo-resistant phenotype, as for example: i) in the ETMR-FBO model (page 14, lines 299-300): “This suggests that RG-like cells are the best suited to survive chemotherapy-induced cell injury and, therefore, associated with chemotherapy resistance.”; ii) In vivo experiments (page 14, line 311-321): “We observed an enrichment of RG-like cells in etoposide-treated samples compared to vehicle-treated ones (4.6 % versus 15.9 %, respectively), supporting the hypothesis that RG-like cells are linked to chemo-resistance (Fig. 6D). To test this hypothesis, we focused on the well-known property of etoposide to induce apoptosis in cancer cells ⁵⁴. We analyzed our dataset for a molecular signature associated with genes that enable cells to recover from apoptotic stimuli ⁵⁵. Our findings revealed that RG-like cells exhibited the highest enrichment of apoptosis-recovery genes (Fig. 6E), which were significantly higher expressed in post-etoposide-treated samples compared to pre-treated ones (Supplementary Fig. S19G). This suggests that RG-like cells

have developed adaptive responses to treatment, supporting them as drug-resistant tumor subpopulations.”

- 4) “Since you are seeing PDGFR signaling as important in both the non-tumor (found to be linked to the RG to PC differentiation that occurs in the presence of tumor cells) and tumor cells, please point that out to signal this at key points in the transitions in the writing about the later figures. “Based on the previous observations” (line 343) is not sufficient. It is worth spending time to be more specific to describe the results and the logic. “hETMR-FBO treated PDGFR inhibitor dataset” = “hETMR-FBO treated with CP-673451”

Response: To strengthen the connection between our prior experiments, findings and the PDGFR treatment performed in the study, we modified lines 334-338, as follows: “Our findings established that PC are central to the ETMR microenvironment. PC are recruited into the TME via PDGFR signaling in ETMR-FBO, engage in stem cell signaling with tumor cells, and exhibit PDGFR signaling patterns associated with chemo-resistant RG-like cells. We therefore evaluated the PDGFR inhibitor CP-673451 for its potential to reduce ETMR cell survival both in vitro and in vivo.” In addition, we substituted the unspecific term “PDGFR inhibitor” for the specific term “CP-673451” throughout the related manuscript section.

- 5) “Additional clarity in the text will be useful – for instance, please discriminate more clearly which systems are human v. mouse (2D co-culture, FB organoid model) when first introducing them; the addition of the mouse and human (and model) symbols in the figures was very useful.”

Response: We thoroughly redraft the manuscript to enhance clarity regarding the cohorts analysed in the study, making modifications across multiple sections. For example, we significantly updated Fig. 1 and the accompanying supplementary figures to clearly separate results from the ETMR genetically engineered mouse model (Fig. 1B-D, Supplementary Figs. S1-S4) from those of primary human ETMR (Fig. 1E-F, Supplementary Figs. S5-S6). This reconstructing aimed to facilitate the reader’s navigation through the distinct datasets. Moreover, we now clearly indicate transitions when presenting sequential findings from different datasets. For instance, on page 8, lines 158-173: “Firstly, we selected the tumor-specific interactions (interactions present in the mETMR-FB but absent in the mFB)” ... “Secondly, we confirmed the contribution of PC to significant interactions within the tumor-specific categories of ‘ECM,’ ‘Stem cell signaling,’ and ‘Angiogenesis’ in the human dataset (n

= 9 samples) (Fig. 2F).” We believe these revisions significantly improve the manuscript’s organization and readability.

6) “It might be worth considering that the enrichment of the Nprog-like cells in ETMR is not due to a stall (line 113), but rather to enhanced fitness of those cells in the tumor.”

Response: We have now incorporated the suggested considerations and added a conclusive perspective by modifying lines 132-137, as follows: “(...) NProg-like cells were the predominant subpopulation in both murine and human ETMR. The prevalence of this tumor subpopulation points to two distinct hypothesis: either a stalled differentiation in ETMR, supported by its transcriptional regulation pattern, or a proliferative / adaptive advantage of the NProg-like cells. For both hypothesis, the maintenance of a stem cell-like state with self-renewing potential appears to be a crucial survival advantage for the tumor.”

7) “The authors appropriately use the Visium data in a circumspect manner; it would be helpful if they quantify the difference in proliferation (Ki67) between Nprog-like areas v. RG-like and Nb-like areas.”

Response: We have generated new analysis from our spatial transcriptomics data and added the information into the Supplementary Fig. S11F (see below). MKI67 correlates better in spatial transcriptomics with ETMR regions enriched for NProg-like cells.

- 8) “It is not clear from the inferred ligand-receptor interaction analysis that the outgoing PC interactions are mainly to immature (RG, NP) rather than also to mature (NB) cells so it is possibly worth rephrasing that.”

Response: In order to clarify the interpretation about the distribution of PC interactions among ETMR cell subpopulations, we adapted Fig. 2E (see below) revised lines 151-154, page 8, to the following: “Two-thirds of the outgoing signals (ligands) from the PC cells were mainly directed to tumor cells with self-renewing potential (RG-like and NProg-like) in both murine and human microenvironments (Fig. 2E), raising the hypothesis that PCs could be related to stem cell signaling in ETMR.”

- 9) “Also, it would help to more clearly state the reason for the focus on PC; presumably the EC L-R interactions seen in the human data were not followed because of the inability to detect EC in mouse?”

Response: We recognize the need to provide further explanations regarding our focused interest in the PC population. Our single-cell transcriptomic analysis of primary murine and human ETMR tumors highlighted a potential role of PCs in stem cell signalling, a hallmark of ETMR biology. Building on this observation, we pursued the hypothesis that PC might contribute to this unique and fundamental characteristic of ETMR. Understanding this contribution could provide critical insights into the mechanisms underlying the aggressive phenotype of ETMR. We detailed this hypothesis in lines 155-158: “As stem cell signaling is a hallmark of ETMR molecular biology, we followed the hypothesis that PC might contribute to this fundamental characteristic of the tumor. Aiming to understand the underlying mechanisms, we investigated the biological role of interactions between PCs and the tumor cells.”

Regarding the reviewer’s question about the role of endothelial cells (EC) in the ligand-receptor (L_R) interaction analysis, the lack of these cells in the single-cell datasets of both murine and human ETMR prevented us from drawing conclusions that could be validated across distinct cohorts.

10)“Please list in the text the quantification of the increase in NP cells shown in Fig. S11F and statistical analysis (stats needed elsewhere too such as Figure 6C and 6D). It will be helpful to know if this increase in NP cells following co-culture with PC cells has been repeated and is a robust and reproducible result.”

Response: We thank the reviewer’s observation about the mentioned result. Since the data previously shown in Supplementary Figure S11F did not provide substantial support for the hypothesis and lacked statistical significance, we have decided to withdraw it. However, we retained the data presented in Figure 7C-D of the revised manuscript (relative to the previous Figure 6C-D) due to its relevance and have now included non-significance statistics to provide a more comprehensive representation.

11)“Can the authors address the difference in CD13⁺ PC between hyper and hypocellular parts of ETMR; does the detected difference simply reflect a higher density of vessels in hypercellular regions, or are the vessels in the hypercellular regions more likely to have pericytes than the vessels in hypocellular areas of the tumor?”

Response: We addressed the request regarding the spatial relationship between tumor subpopulations and pericytes by performing multiplex immunofluorescence imaging. This analysis revealed not only the expected association of pericytes with blood vessels within the tumor but also demonstrated a distinct proximity between self-renewing tumor cells (RG-like, NProg-like) and perivascular regions enriched with pericytes. We presented these results in a new Fig. 3A-C of the revised manuscript (see below). PDGFR β ⁺ PC co-localize with the smooth-muscle marker ACTA2 and the EC marker CD31, juxtaposing EC in vessel walls (left panel). In addition, hypercellular areas enriched in RG-like (NES^{high}/SOX2⁺) and NProg-like (SOX2⁺/NES^{low-}) cells develop around PDGFR β ⁺ vascular regions (right panel). Moreover, quantitative analysis validate a significant proximity of RG-like and NProg-like cells to PC regions in comparison to Nb-like (MAP2⁺) cells (density plot).

12)“It is not clear what effect co-culture with FBO derived from hiPSC has on the identity of the human BT-183 and mouse GEMM-derived EMTR tumor cells which they are co-cultured with. How do the fractions of RG, NP, and NB-like cells change from before and after co-culture? Maybe I missed it.”

Response: We apologize for the lack of clarity in presenting this data. This information is available in Fig. 3H. To enhance clarity, we have updated the figure to include the percentages for each cell type across all experimental models, including murine primary, human primary, 3D organoids and 2D culture.

13)“Changes would help to the abstract and to the discussion to clearly list the identities of the three distinct ETMR tumor cell populations (RG-like, NProg-like, and Nb-like).”

Response: We responded to the reviewer’s request by naming the tumor subpopulations explicitly in: i) lines 14-15 (Abstract): “We revealed three distinct malignant ETMR subpopulations in a putative neurodevelopmental hierarchy (RG-like, NProg-like and NB-like) (...);” and ii) lines 364-366 (Discussion): “In this study, we describe ETMR cellular states as

three distinct tumor cell subpopulations (RG-like, NProg-like and NB-like) characterized by particular transcriptional, metabolic, and cell cycle programs.”

14)“A schematic summarizing the author’s model of tumor and vascular populations and their crosstalk would be useful.”

Response: We highly valued the reviewer’s suggestion. We, therefore, created a new figure (Fig.8, revised manuscript) showing a schematic overview of the main findings in this manuscript in a comprehensive graphical summary.

15)“While it is helpful to have much of the supporting data in the supplementary figures, it might help to move some additional key aspects of the study to the main figures (perhaps a new Figure 2 that contains elements of the supplemental figures 4-8).”

Response: We thank the reviewer for acknowledging the richness of the data presented in the previous Supplementary Figures 4–8. A detailed description of the developmental hierarchy of tumor subpopulations, their differences in cell metabolism, cell cycle, signalling pathways, and transcriptional regulation is provided in these figures (now Supplementary Figs. 2–5).

However, to maintain the manuscript's focus on the relationship between tumor cell populations, pericytes, and the mechanisms of chemo-resistance, we have opted to keep this detailed description in the supplementary materials. This approach ensures a clear and cohesive line of argumentation throughout the main text.

That said, if the reviewer board considers these points sufficiently significant, we would be happy to create a new main figure to incorporate and highlight these results.

Reviewer 4:

“The paper “ETMR stem-like state and chemoresistance are supported by perivascular cells at single-cell resolution” by de Faria et al., is an important paper that captures the cellular heterogeneity of ETMRs at exquisite detail, using a combination of primary tumor samples, organoid models and xenografts. The paper highlights the crucial role of the TME in ETMRs and challenges the current paradigm which often posits ETMRs as being immunologically cold. While this is a novel finding, the study suffers from biologically relevant mechanistic insights that would help further therapeutic approaches, as detailed below –

Response: We sincerely appreciate the reviewer's positive feedback on our manuscript, particularly regarding its scientific relevance, detailed descriptions, cross-validations, and novelty. We apologize for the lack of validation in certain mechanistic insights and hope that the revisions and additional analyses we have provided address the reviewers' concerns in an adequate and thorough manner.

1) The existing literature around ETMRs show conclusively that the oncogenic drivers of ETMRs are an upregulation of C19MC amplification, coupled with DICER1 mutations that lead to a dysregulation of miRNA's. The current study, shows a new role of oncogenesis in ETMRs – one that involves the WNT-PDGF signaling cascade. However, the novel pathway alluded in this study, fails to connect to the existing literature on C19MC induced ETMR oncogenesis. It would be helpful if the authors can address whether such a link exists and how the link can be manipulated with PDGFR inhibition. This is a crucial question, as much of the clinical decision making, in terms of ETMR recurrence stems from an evaluation of the C19MC status. If a patient fails conventional therapy, as they often do, would selective PDGFR inhibition play a role in stemming cancer growth? Also, which of the 3 sub-populations of ETMRs house the C19MC cells? This is also relevant as much of the current strategies are aimed at eliminating these oncogenic cells. Without a mechanistic link to C19MC, the paper lacks potential therapeutic strategies that are clinically relevant.”

Response: We consider of high relevance the comments raised by the reviewer. To respond to question regarding the interplay between PDGF signalling and ETMR molecular biology, we have added relevant information about the topic in the session “Discussion”, as follows in page 17, lines 390-397: “Notably, there is an interplay between PDGFR signaling and ETMR biology. Let-7 family miRNAs is suppressed in ETMR, leading to the upregulation of LIN28A, a molecular hallmark and a clinical marker of this tumor type ⁶. Conversely, Let-7 miRNAs function as upstream regulators of PDGF signaling. There is in vitro evidence for the increased expression of PDGF and its receptor (PDGFR) in smooth muscle and endothelial cells upon suppression of let-7. Distinct cell lines treated with PDGF showed let-7 reduction, partially rescued by imatinib, a PDGF pathway inhibitor ⁵⁶. In the clinical setting, a second-generation PDGF inhibitor, dasatinib, induced tumor response and long-term survival of a child with refractory ETMR ⁷².”

We understand the significance of the reviewer’s insightful question about the relationship between C19MC, the ETMR subpopulations found in our study and tumor recurrence. However, this topic falls outside the scope of our study for two primary reasons: Firstly, the single-cell technologies employed in our research are not suitable for investigating miRNAs due to intrinsic technical limitations; specifically, oligonucleotide fragments less than 200 bp, which include miRNAs, are systematically excluded during quality control steps in sample preparation for sequencing. Second, examining C19MC in recurrent tumors would necessitate

a systematic analysis of recurrent tissues and/or the development of a chemotherapy-resistant cell line, which was not addressed in our current research.

- 2) “The current study makes use of several bioinformatics strategies to deduce link to VEGF, TGF, PDGF, apoptosis, etc signaling. However, they lack any biochemical or cell biological validation. Over-expression of genes does not imply activation of these pathways. The latter can be validated using phospho-specific antibodies (on Westerns, or IHC) that specifically addresses whether the pathway is active in ETMR cell-lines.”

We agree that validations on different levels are essential to proof important point of a story. Therefore, in this study we already used several dimensions of validation: e. g. i.) cross-model validations (primary human samples, a mouse model and a novel organoid model) and ii.) cross-technical validations (scRNAseq, spatial transcriptomics and multiplex immunohistochemistry). We included new data to stain for different cell types. In conclusion, we believe that we have really created a story with multiple data sets and already validated major findings with several different methods.

- 3) “The authors mention a large component of stem cell signaling in ETMRs between pericytes and tumor cells, but then point to pericytes driving a neurogenic-like proliferative state. As neuronal progenitors are not stem-like, these two statements seem to be at odds.”

Response: We apologize for the lack of clarity when describing cells from distinct developmental states. To address this, we have revised the nomenclature of the cell subpopulations throughout the manuscript to eliminate ambiguity. For example:

- i) We replaced the term “undifferentiated” in the previous abstract (line 18): “PDGF signaling was upregulated in chemotherapy-resistant undifferentiated ETMR cell subpopulations *in vivo*” with “RG-like cells”, clearly specifying the ETMR subpopulation to which the finding pertains. Similar ambiguous terms have been excluded or clarified throughout the manuscript;
- ii) The term “neurogenesis” was removed when its usage did not align conceptually with the idea of neuro-differentiation. For instance, on page 9, lines 184-185: “Among the co-culture unique interactions, we confirmed the presence of L_R pairs associated with signaling in pluripotent stem cells and ECM network, besides

neurotrophic signaling (Wnt, Hippo, neurotrophin) also associated to stem cell niches³⁷ the term was adjusted to ensure coherence.

We have implemented these changes to improve clarity and consistency of the manuscript.

- 4) “The authors demonstrate a reduction of PC cells when treated with PDGFR inhibitor CP-673451. However, I question if a reduction from 0.44% of the TME to 0.04% of the TME serves as a biologically significant change.”

Response: The reviewer raised concerns about the results presented in current Figures 7C-D (revised manuscript). While these results are not statistically significant, we chose to retain them due to their biological relevance to the study. To address this, we have now included statistical analysis for Figures 7C and 7D and revised the text on lines 342-345, page 15, to provide greater clarity regarding our conclusions. The updated text reads: “PDGFR inhibition with CP-673451 resulted in a 10-fold decrease of the PC relative proportions in the TME, although not statistically significant (median: 0.44 % in DMSO-treated versus 0.04 % in CP-treated samples, p-value = 0.42) (Fig. 7C).”

- 5) “Furthermore, we would not expect pericytes in organoid models, but treating the organoid models with PDGFR inhibitors leads to reduction in tumors cells and proliferative Ki-67 positive tumors cells, reinforcing the author’s claims that PDGFR is important in ETMR growth but weakens their own claims that pericytes play a specific role in PDGFR signaling.”

Response: Thank you for your question regarding the presence of pericyte-like cells in the FBOs. The reviewer is correct that PC are not expected in this tissue model, as we demonstrated in Figures 5A, 5B and Supplementary Fig. S15C. Instead, radial glia cells with pericyte-like gene expression develop when the FBOs are co-aggregated with tumor cells, as described in Figure 5D. We revised the manuscript to clarify this point, as follows (page 12, lines 248-255): “In the mETMR-FBO, two unique RG-FBO clusters, termed RG-PC and RG-angio, emerged based on their association with either ECM organization – resembling PC GO terms (RG-PC), or blood vessel development (RG-angio) (Supplementary Fig. S7B, S14E, S15A-B). These novel RG subpopulations, while not expected in FBOs⁴², were present in tumor-co-aggregated FBOs (Fig. 5A-B, Supplementary Fig. S15C). Notably, the emergence of PC-like gene expression in RG-PC cells suggests a tumor-induced lineage shift toward a pericyte-related state.”

Consequently, when ETMR-FBO were treated with the PDGFR inhibitor CP-673451, pericyte-like cells were also available as targets. Additionally, in response to point 1 from Reviewer 3, we have included a figure addressing the same topic for further clarification.

6) “Authors claim EMT in Figure 4E but do not look into canonical EMT transcription factors such as ZEB, TWIST and SNAI.”

Response: Thank you for this comment. The reviewer raised a point regarding the revised manuscript lines 267-269. We did not intend to imply the occurrence of an EMT process, nor is this process mentioned elsewhere in the manuscript. Rather, we referenced prior studies (references 46-48) that associate the genes upregulated in the mesenchymal (*MES*) program with the EMT process, thereby supporting the mesenchymal identity of this program.

7) “I would like to see the enrichment of the PC signatures on the 10X spatial data, to further validate the presence of pericytes.”

Response: We highly appreciate the reviewer’s insightful comment. To address this point, we have integrated the spatial transcriptomic data (n samples = 3), annotated the ETMR spatial regions into RG-like, NProg-like and Nb-like enriched regions based on single-cell deconvolution analysis and performed correlation analysis of PC and EC gene expression with the distinct ETMR spatial regions. We found both PC and EC better correlated with RG-like enriched tumor regions. The data generated from this analysis is now included in the Supplementary Figure S11G of the revised manuscript (see below).

- 8) “I would also like see some form of spatial organizational analysis to investigate cell-cell signaling with the spatial information integrated. I would also ask if spatial distribution of pericytes is significant, or if a random spatially distributed pericyte population would lead to similar significance.”

Response: Coherent with the previous question, the spatial transcriptomics data presented in the Supplementary Fig. S11G reveal an association between pericytes and tumor regions that are particularly enriched in RG-like cells. To further investigate the spatial relationships of PC within the ETMR TME at protein level, we performed multiplex immunohistochemistry on a human FFPE sample (n = 1, Fig. 3). Our results demonstrate that pericytes are localized in vascular areas of the tumor, as expected. These regions are consistently surrounded by NES⁺ and SOX2⁺ tumor cells, representing RG-like (NES^{high}/SOX2⁺) and NProg-like (SOX2⁺/NES^{low}) tumor cells. We included a figure representing these results in response to Reviewer 3, point 11.

- 9) “The authors use the data presented in Suppl figure 8 to validate the transcriptomic heterogeneity of 3 primary ETMRs. However, there is heterogeneity among the three ETMRs to make any meaningful conclusions. For example, the ETMR signature correlates strongly with Lin28A in T2, but not with Ki-67, and even less with NProg-like signature or Nb-like signature. In T1, there are multiple populations with RG-like, Nprog-like and Nb-like signatures, which does not really correspond to the ETMR signature.”

Response: We apologize for the lack of clarity in the initial interpretation of the spatial results. We recognize that the original presentation of Supplementary Figure S8A may not have been sufficiently conclusive. To address this, we re-analysed the spatial data using a bioinformatic deconvolution approach, which allowed for a more precise alignment of our single-cell data with the spatial resolution. The updated data is now represented in a new format, providing quantitative results that we hope to address the reviewer’s concerns.

The Supplementary Fig. S8A, now Supplementary Fig. S11E, illustrates the spatial distribution of deconvoluted RG-like, NProg-like and Nb-like enriched regions. These regions, while occupying distinct tumor areas, together represent nearly the entire gene expression profile of the tissues analysed. Given the heterogeneity of the cell subpopulations, a unified gene expression signature does not fully capture the tissue complexity, and thus we have excluded this signature from the results.

Additionally, we integrated the spatial data (n samples = 3) to provide a quantitative and more comprehensive interpretation for the correlation of LIN28 and MKI67 across samples. These

results are presented in the revised Supplementary Fig. S11F, where it is shown that LIN28A and KI67 primarily localize to regions enriched with NProg-like cells. The reviewer can also refer to the figure in response to Reviewer 3, point 7, for further context.

10)“In Figure 2E, the authors state that PC-ETMR signatures are “mainly” to Rg-like and N-prog-like population. Judging from the pie-chart, in the absence of any provided numbers, one would conclude that the interactions are fairly similar between PC and three ETMR sub-populations.”

Response: To respond to this request, we modified the lines 151-153, page 7, as follows: “Two-thirds of the outgoing signals (ligands) from the PC cells were mainly directed to tumor cells with self-renewing potential (RG-like and NProg-like) in both murine and human microenvironments (Fig. 2E), raising the hypothesis that PCs could be related to stem cell signaling in ETMR.” In addition, we provide now the relative proportions embedded in Figure 2E.

11)“In Figure 2G, the contributions of NB-like cells interactome with PC cells and EC cells are seemingly ignored, with the focus shifting primarily to RG-like and Nprog-like cells. How do the authors determine significance of interactions, is it purely subjective?”

Response: Thanks for raising this conceptual question. We didn't find a meaningful number of endothelial cells in the scRNA-seq datasets (murine or human), which prevented us from any meaningful conclusions involving this cell type, as we would lack datasets for appropriate validations. In Figure 2G, we presented all the significant ligand-receptor interactions predicted by CellphoneDB tool, unbiasedly selecting those associated with “ECM”, “Stem cell signalling” or “Angiogenesis”, as these are components of tumor-specific signalling pathways (see Figure 2F). Since our primary interest was in the role of pericytes in tumor-related stem cell signalling, we focused on PC interactions with the tumor cells harboring self-renewing potential (RG-like and NProg-like cells). In Figure 2G, while NB-like cells contribute to all the tumor-specific signaling, they do not interact with pericytes for stem cell signalling, rather with RG-like and NProg-like cells, which supports the hypothesis of pericytes contributing to tumor stemness. We have revised the manuscript (page 8, lines 171-175) to better clarify the idea: “Secondly, we confirmed the contribution of PC for significant interactions within the tumor-specific categories “ECM”, “Stem cell signaling” and “Angiogenesis” in the human dataset (n = 9 samples) (Fig. 2F). Although all the tumor subpopulations participate in the three categories of tumor-specific signaling, PC engage only with RG-like and NProg-like tumor subpopulations for the class “stem cell signaling”.”

12)“Supplementary figure 11F – Based on the figure and the accompanying legend, with a p value of >0.05, and with the large error bars in mono-culture, one would conclusively state that the pericytes do not change the proportions of NP-Prog like cells. However, the authors conclude that pericytes drive tumor cells to a neurogenic-like proliferative state. On what basis is this conclusion made?”

Response: We acknowledge the reviewers’ concern regarding the representability of the result in Supplementary Figure S11F. Since this result does not add substantial support for the hypothesis and lack statistical significance, we have withdrawn it from the manuscript.

13)“In Supplemental figure 11 D – Stmn4 is replaced by Stmn2 between the mono-culture vs. co-culture conditions, precluding one to make a conclusion about the differences in expression between the stemness genes in different sub-populations.”

Response: We apologize for the mistake in current Supplementary Fig. S9D-E (revised manuscript). We modified it to represent the same gene in both conditions. Now Stmn4 expression is described for both mono- and co-culture conditions.

14)“In Suppl figure 13 – the authors conclude that pericytes co-localize with hyper-cellular regions of the ETMR. Do they see a similar co-location with endothelial cells – In other words, is the co-localization of pericytes in the hyper-cellular areas due to the enrichment of blood vessels in the hyper-cellular zones?”

Response: We addressed before (point 8) the reviewer’s request regarding the spatial relationship between tumor regions and pericytes by performing multiplex immunofluorescence imaging. Recapitulating briefly, this analysis revealed not only the expected association of pericytes with endothelial cells within the tumor but also demonstrated a distinct proximity between hypercellular regions enriched in self-renewing tumor cells (RG-like, NProg-like) and pericyte-positive blood vessels.

15)“In Figure 5D – There is a decrease in RG-like signatures in Day 30 untreated vs Day 48 untreated controls. Do the authors have any explanations of why the RG-like population fluctuate between the two controls, and how this can potentially affect their interpretation of the data.”

Response: We appreciate the reviewer’s insightful observation. The referenced Figure 5, now updated as Figure 6 in the revised manuscript, presents data from ETMR-FBO models treated

with carboplatin or etoposide and harvested at either day 30 (D30) or day 48 (D48) of organoid culture for scRNA-seq analysis. As brain organoids develop and mature over time, it is plausible that a more developed brain environment may create unfavorable conditions for tumor cells with a stem cell-like phenotype, such as RG-like tumor cells. Although this remains speculative, it is noteworthy that ETMR almost exclusively arises in very young children, where the brain is still immature.

Acknowledging the reviewer's valid concerns regarding timepoint comparisons, we re-analysed the data and now ensure that all comparisons are made exclusively with controls from the corresponding timepoint. This adjustment aims to address the reviewer's concerns and enhance the rigor of our analysis.

16)“In Figure 5D – what happens to the Nb-like cells following treatment with Etoposide/Cisplatin?”

Response: We apologize for the lack of information about Nb-like cells in this result. As the figure was already complex and containing multiple panels, we chose before to withdraw this additional result. However, in line with the prior response and also observing the request from another reviewer, we significantly modified the results presented in the revised Fig. 6. The revised figure is substantially more concise and now accommodate the findings of the Nb-like cells. The comments on the modification of this figure and a representation of the figure itself can be found in response to Reviewer 3, point 2.

17)“Using tSNE could confound how transcriptionally distinct or not the clusters are. I would like to see UMAP reduction in Figure 1 to be more up to date.”

Response: Even though we appreciate the reviewer's insights on the differences between UMAP and tSNE data representations, we would like to emphasize that neither approach alters the actual transcriptional distinction between cells; they only influence how the data are visualized and clustered. tSNE is particularly effective at preserving local structure and distinguishing clusters in high-dimensional data, making it valuable when it is crucial to separate cells from different sample origins. Unlike UMAP, tSNE tends to maintain these distinctions without blending clusters as readily, which is especially useful when there are no clear markers, such as copy number variations, to separate cell types or populations. Since we did not expect copy number variations in the murine data to help identify malignant cells, tSNE clustering prevents the mixing of tumor and non-tumor cell populations, allowing for more reliable comparisons with the control murine dataset. Therefore, we have chosen to retain the murine data in tSNE format. Please refer to Marx, V. Seeing data as t-SNE and UMAP do. Nat

Methods 21, 930–933 (2024). <https://doi.org/10.1038/s41592-024-02301-x> further information on the topic.

18)“It is unclear what samples contribute to the pericyte populations in Figure 1B, as cluster numbers are missing from the tSNE”

Response: To accommodate the reviewer’s request, we modified Figure 1B and 1C, which describes now the distribution of the distinct cell populations by cell type, aligned with the annotation described in Figure 1C.

19)Technical documentation could be more detailed as to batch corrections on which combinations of datasets

Response: To clarify the batch correction approach, we revised the methods section of the manuscript and added the following in lines 850-853, page 37 (Methods): “The sample composition for every integration described in the manuscript is presented in the supplementary table 3a. The batch correction was performed for all integrations on a sample level and using the anchor-based integration method from Seurat.”

20)I wonder if integrating the FFPE samples alongside the fresh frozen human samples would provide additional insights into the heterogeneity of ETMRs, or would the technologies be too distinct for integration.

Response: We thank the reviewer for this relevant consideration. The sample preparation methods for fresh frozen and FFPE specimens differ substantially, as they rely on distinct chemistries (reverse transcription-based for fresh frozen samples and probe-based for FFPE samples). Consequently, combining data from both types is expected to introduce significant batch effects, as the count matrix will certainly differ. Furthermore, one of the fresh frozen human samples was processed in a different laboratory, adding another source of batch variability. Given that we had only two fresh frozen samples, we chose not to compromise the integrity of the FFPE RNA data by forcing integration with the fresh frozen samples. Instead, we used the fresh frozen samples as an independent validation cohort.

21)Please italicize “in vitro” and “in vivo” throughout the manuscript.

Response: We appreciate the reviewer comment on the manuscript formatting for consistency. The use of italics for expressions like “in vitro” and “in vivo” seems to be controversial, as pointed in the online format of NCBI Style Guide available at <https://www.ncbi.nlm.nih.gov/books/NBK995/>. Although they recommend the use of italics, the

Council of Science Editors (CSE manual) recommend not to use it. As it is a matter of style, we will let the editorial board of *Nature Communications* to liberate about this point further in the revision process.

Point-by-point responses to referee comments - second revision

We are deeply honoured by your expression to publish our manuscript “**ETMR stem-like state and chemo-resistance are supported by perivascular cells at single-cell resolution**” in *Nature Communications* after addressing remaining points of reviewer 5. We sincerely thank the reviewers for their thorough and thoughtful evaluations, which have greatly contributed to strengthening the clarity, consistency, and overall quality of our work. Their comments and suggestions not only improved the readability of the manuscript but also played a key role in shaping a successful further submission.

We approached the remaining points raised during the review process with dedication. In this response, we provide detailed answers to reviewer 5 comments and summarize below the new data incorporated into the manuscript as a result of the second revision:

New Data	Reviewers point	Figure
Signaling pathways enriched in PC and predicted by GSEA molecular pathways	Reviewer 5 point 7	Supplementary Fig. S7C
Description of ligand-receptor pairs (L_R) in interactions between PCs and ETMR cell populations	Reviewer 5 point 8	Fig. 2F- 2G
Spatial resolution of the cross-talk between PC and ETMR cell populations described in Figure 2F- 2G.	Reviewer 5 point 8	Supplementary Fig. S11H

Please find below the point-by-point answer to the reviewers’ comments.

Reviewer 1:

“The authors have fully addressed all of my comments. I congratulate the authors for this important work and endorse publication.”

Response: We thank the reviewer’s positive feedback on our work and their collaboration for the improvement of this manuscript.

Reviewer 2:

“My comments have been adequately addressed.”

Response: We are glad to have successfully addressed reviewer’s 2 comments.

Reviewer 3:

“The authors have addressed my comments by substantially improving the clarity of the figures and the text – particularly regarding terminology and experimental setups. The additional quantification is useful, and the schematic is another valuable addition to help readers understand the rather detailed aspects of this manuscript. I understand the authors’ point against relocating the detailed supplementary data. I think it would help to add a sentence regarding the rationale for not pursuing endothelial cell L-R interactions in EMTR.”

Response: We appreciate the reviewer’s careful attention to our manuscript and their many valuable suggestions. In response to the comment regarding our rationale of not investigating ligand-receptor (L-R) interactions involving endothelial cells, we have added the following explanation to the revised manuscript (lines 381-384, page 17): “Unfortunately, the lack of endothelial cells in our scRNA-seq datasets, likely due to our sample preparation approach, precluded further investigation into how L-R interactions with PCs may have influenced their gene expression and recruitment to the TME.”

Reviewer 5 (replacing Reviewer 4):

“The paper has been modified based on the reviewers suggestions and the addition of clarifying data and statements makes the paper easier to read.

- 1) However, the paper lacks mechanistic insight, which I feel is the main drawback of the paper. It fails to draw a link to the current dogma of ETMRs being driven by C19MC/Dicer1 mutations. While I understand the technical limitations of not being able to detect miRNA in single cell technologies, Lin28 can potentially be used as a surrogate marker. Jessa et al (Nature genetics 2019) used TTYH1 as a surrogate marker. Furthermore, the addition of the C19MC pre-mRNA as a pseudogene has been demonstrated to be a robust readout of C19MC activity. (also same reference).”

Response: We sincerely appreciate the reviewer’s positive feedback on our manuscript and we apologize for the lack of validation regarding certain mechanistic insights. Reviewer 4 previously requested clarification on the relationship between C19MC activity in ETMR and our investigation on PDGFR signaling. We were aware of the use of C19MC pre-mRNA as a

pseudogene by Jessa et al., 2019 and we have previously attempted to align a correspondent sequence to the human genome. However, we were not successful. We couldn't find further details within their manuscript regarding which specific sequence was used for the alignment or how they have identified corresponding reads in the data (as pre-mRNAs are normally not poly-A tailed, therefore not captured by 10x genomics chemistry), which prevented us to reproduce their findings.

Regarding LIN28A expression, we have already included information on its spatial expression in our data in the first revision (Supplementary Figure S11F). In our snRNA-seq dataset (n = 9), LIN28A exhibits low expression, consistent with the findings of Jessa et al., 2019. Similarly, as previously reported by the same authors, TTYH1 is expressed predominantly by tumor cells, but also by non-malignant cell types. Nevertheless, we are pleased to provide requested data on both LIN28A and TTYH1 expression (see below).

That said, confirming of the expression of these genes does not directly address the question regarding their mechanistic relationship with PDGFR signaling. Our data demonstrated the upregulation of PDGFR signaling in chemoresistant murine ETMR RG-like cells compared to untreated ones, which may reflect a stress-induced cell response rather than an effect driven by the intrinsic genetic background of the tumor. Addressing this fundamental question would require a dedicated study involving the generation of a chemoresistant ETMR cell line and gene knockout approaches to assess gene dependencies – efforts that fall outside the scope of the current work.

To acknowledge this limitation, we have added the following text to the revised manuscript (lines 412-416, page 18): “Even though our study demonstrated the upregulation of PDGFR signaling in chemoresistant murine ETMR RG-like cells compared to untreated controls in silico, we did not generate chemoresistant cell lines to validate this findings in vitro or investigate the interplay between PDGFR signaling and ETMR genetic alterations. It remains unclear whether PDGFR upregulation is a stress-induced response or intrinsically related to the ETMR genetic background – representing a limitation of our study.”

2) “None of the assays that have been used in this study are functional. The authors have access to ETMR cell lines (BT183). Analyzing the functional status of the PDGF pathway and/or analyzing the survival of these cells in face of PDGFR inhibition would lend more credence to the main assertions of this study.”

Response: We demonstrated that ETMR cells surviving in vivo etoposide treatment upregulated PDGFR signaling as a potential survival mechanism, and that treatment of embryos with the PDGFR inhibitor CP-673451 leads to reduced KI67 expression in tumor regions. Although we do not expect PDGFR signaling upregulation in non-chemoresistant cells, we were unable to perform phospho-protein immunohistochemistry analyses to clarify this due to time and technical constrains. Therefore, we acknowledged this as a limitation of our study and included the relevant information in the revised manuscript (lines 412-416, page 18), as also mentioned in our response to the previous question.

Comments 3 to 6: “Thank you for making these changes.”

Response: We appreciate to have satisfied the reviewer’s request with our answers to that specific points.

7) “This analysis suggests that indeed the PC signature is tumor-related, but then it is somewhat in disagreement with the previous result, where it states that TME (non-malignant) PCs are what drive signaling. I would like some clarification on if this PC signaling is tumor or TME related.”

Response: We thank the reviewer for this relevant question. We have previously provided CNV analysis from our human scRNA-seq cohort (n = 2) and from fixedRNA cohort (n = 9) in the Supplementary Figures 5D and 6D, respectively. We have now updated Supplementary Figure 6D to display the cell types associated with the reference clusters. As shown (see below), pericytes do not harbour chromosome 2 gain, a genomic alteration frequently observed in over 70% in ETMR tumors (Korschunov et al., 2014). Based on this observation, we classified pericytes as originating from the tumor microenvironment.

8) “While the pericyte signature is enriched within the tumor mass on a spatial level, it still isn’t clear if this cell-cell signaling is occurring. I would like to see some plot of L-R signaling as predicted by single cell to validate that cells are indeed expressing these LR genes, and that they are closely spatially organized supplement the multiplex IHC.”

Response: We appreciate the reviewer’s observation regarding the lack of information on specific ligand-receptor interactions between pericytes and ETMR cells. To address this point, we have added to Figure 2 the predicted ligand-receptor pairs for each biological category (ECM, Stem cell Signaling and Angiogenesis) identified in our snRNA-seq dataset. Additionally, we performed spatial correlation analysis of the same ligand-receptor pairs using our human spatial transcriptomics data and included this information as Supplementary Figure S11H (see below). The complementary analysis support that cell-cell communication between pericytes and tumor cells is primarily mediated by extracellular matrix components.

We thank the reviewers once again for their thorough evaluation and constructive feedback, which have greatly improved the quality and clarity of our manuscript. We believe that the revisions made in response to the current round of comments have further strengthened the study, and we hope that it is now suitable for publication. We remain grateful for the time and effort invested in reviewing our work and are happy to provide any additional information if needed.

With our best regards,

Flavia W. de Faria and Kornelius Kerl